# Interactions between nocturnal turbulent flux, storage and advection at an 'ideal' eucalypt woodland site

Ian D. McHugh[1], Jason Beringer[2], Shaun C. Cunningham[3,6], Patrick J. Baker[4], Timothy R. Cavagnaro[5], Ralph Mac Nally[6], Ross M. Thompson[6]

[1]School of Earth, Atmosphere and Environment, Monash University, Melbourne, 3800, Australia
[2]School of Earth and Environment, University of Western Australia, Perth, 6907, Australia
[3]School of Life and Environmental Sciences, Deakin University, Melbourne, 3125, Australia
[4]Forest dynamics laboratory, University of Melbourne, Melbourne, 3052, Australia
[5]Waite Research Institute and School of Agriculture, Food and Wine, University of Adelaide, Adelaide, 5005, Australia
[6]Institute for Applied Ecology, University of Canberra, Canberra, 2617, Australia

*Correspondence to*: Ian D. McHugh (ian.mchugh@monash.edu)

**Abstract.**

While the eddy covariance technique has become an important technique for estimating long-term ecosystem carbon balance, under certain conditions the measured turbulent flux of $CO_2$ at a given height above an ecosystem does not represent the true surface flux. Profile systems have been deployed to measure periodic storage of $CO_2$ below the measurement height, but have not been widely adopted. This is most likely due to the additional expense and complexity, and possibly also the perception – given that net storage over intervals exceeding 24 hours is generally negligible – that these measurements are not particularly important. In this study, we used a three-year record of net ecosystem exchange of $CO_2$ and simultaneous measurements of $CO_2$ storage to ascertain the relative contributions of turbulent $CO_2$ flux, storage and advection (calculated as a residual quantity) to the nocturnal $CO_2$ balance, and to quantify the effect of neglecting storage. The conditions at the site are in relative terms highly favourable for eddy covariance measurements, yet we found a substantial contribution (~40%) of advection to nocturnal turbulent flux underestimation. The most likely mechanism for advection is cooling-induced drainage flows, the effects of which were observed in the storage measurements. The remaining ~60% of flux underestimation was due to storage of $CO_2$. We also showed that substantial underestimation of carbon uptake (approximately 80 gC $m^{-2}$ $a^{-1}$, or 25% of annual carbon uptake) arose when standard methods (u* filtering) of nocturnal flux correction were implemented in the absence of storage estimates. These biases were reduced to approximately 40-45 gC $m^{-2}$ $a^{-1}$ when the filter was applied over the entire diel period, but they were nonetheless large relative to quantifiable uncertainties in the data. Neglect of storage also distorted the relationships between the $CO_2$ exchange processes (respiration and photosynthesis) and their key controls (light and temperature, respectively). We conclude that addition of storage measurements to eddy covariance sites with all but the lowest measurement heights should be a high priority for the flux measurement community.

# 1 Introduction

Over the past 2 decades, eddy covariance measurements have been widely adopted as a tool for aggregate flux measurement (Baldocchi, 2003), and there are now over 650 operational monitoring sites registered with the international flux network (Fluxnet: fluxnet.ornl.gov). Within the Australian regional network (OzFlux: www.ozflux.org.au), there are 29 active sites (Beringer et al., 2016, this issue). The use of the eddy covariance technique allows continuous automated monitoring of mass and energy fluxes, and long-term multi-site datasets have yielded valuable ecological insights in recent years (Baldocchi, 2008).

It has long been documented that eddy covariance measurements are prone to underestimation of the true surface efflux of $CO_2$ at night (e.g. Goulden et al., 1996). The key processes associated with this underestimation are storage and advection (Aubinet et al., 2012). In the former process, $CO_2$ may be stored below the measurement height under calm conditions. However, since the $CO_2$ mole fraction must be approximately preserved over longer time scales (e.g. 24 hours +), $CO_2$ respired and accumulated nocturnally is generally released with the initiation of buoyancy-generated turbulence following sunrise, such that net storage is zero.

Advection involves mean transfer of $CO_2$ due to the development of horizontal and vertical gradients in scalar fields; the primary process by which this occurs is the initiation of gravity-driven drainage currents on sloping terrain (Aubinet, 2008). In contrast to storage, this mechanism generally results in a net loss of $CO_2$ from the observing system. Since substantial respiration occurs nocturnally, this loss causes a selective systematic error towards $CO_2$ uptake for cumulative carbon budgets at time scales > 24 hours. These drainage flows have been observed to occur on slopes of < 1° (Aubinet et al., 2003; Staebler and Fitzjarrald, 2004). Most non-agricultural ecosystem measurement sites are on sloped terrain because historically, flat, arable land has been cleared for agriculture. Drainage-induced advection is therefore thought to occur to varying extent at most sites.

The storage term can be calculated using relatively simple instrumentation that measures $CO_2$ concentrations along a vertical profile between the eddy covariance instrumentation and the ground (Yang et al., 2007). In contrast, attempts to measure advection have involved deployment of complex instrumentation and delivered highly uncertain results (Aubinet et al., 2010; Leuning et al., 2008). Thus indirect approaches to data correction have been devised, the most common of which is the identification of a threshold below which the nocturnal turbulent $CO_2$ flux declines with turbulent activity (as expressed by friction velocity [$u_*$]) (Goulden et al., 1996). Since there should be no relationship between $u_*$ and ecosystem respiration, this

decline is interpreted as an increase in the storage and advection terms at the expense of the turbulent flux. Data below this threshold (herein $u_{*th}$) are discarded and replaced using functional relationships between known physical respiratory drivers (primarily temperature) and nocturnal $CO_2$ fluxes.

This $u_*$ filtering has been criticised on theoretical and practical grounds (Aubinet et al., 2012; Aubinet and Feigenwinter, 2010; Van Gorsel et al., 2007), but remains the most widely adopted approach to nocturnal data correction. It should be applied to the sum of the measured turbulent flux and storage terms (Papale, 2006), but is quite commonly applied to measurements of turbulent flux in isolation, rather than the sum of the turbulent flux and storage terms. This is because only 10-30% of sites globally have deployed profile systems to measure storage (Papale, pers. comm., 23/11/2015). In Australia,
only four of the 29 active sites have profile measurements, whereas $\geq$ 15 sites have canopies of sufficient height to warrant them (using measurement height > 3 m as a threshold for requirement).

As noted, while the net storage over time is approximately zero, it is generally positive nocturnally as $CO_2$ accumulates below the measurement height. In the morning, the sign reverses due to two processes: 1) the turbulent transfer of the
accumulated $CO_2$ upwards through the measurement plane, and; 2) photosynthetic $CO_2$ uptake by the canopy. Thus neglect of storage means that nocturnal respiratory $CO_2$ release is underestimated, but this is balanced by underestimation of morning ecosystem photosynthetic $CO_2$ uptake.

However, when nocturnal $u_*$ filtering is applied, it implicitly accounts for both storage and advection. Since there is no
corresponding correction for the reversal of the storage term after sunrise, the requirement for the storage term to be approximately zero is violated, and the nocturnally respired $CO_2$ is effectively counted twice (Aubinet et al., 2002). This unavoidably biases measurements towards net $CO_2$ efflux, and also affects the apparent relationship between ecosystem $CO_2$ fluxes and climatic controls.

Given the number of sites that do not have profile systems, it is thus important to quantify the effects of failing to measure storage. In this study, we use a three year-record of $CO_2$ exchange (including storage measurements) for an Australian eucalypt woodland to investigate the interaction between nocturnal turbulent flux, storage and advection. We devise a simple method to infer the magnitude of advection and in turn quantify the apportionment of the nocturnal ecosystem $CO_2$ source between turbulent flux, storage and advection. We quantify the biases in annual carbon exchange that arise from neglecting
the storage term and discuss its effects on interpretation of $CO_2$ fluxes in the context of climatic drivers. Given the significant additional investment and complexity associated with the construction and deployment of profile systems alongside eddy covariance systems, it might be argued that the incurred bias of neglecting storage could be ignored if it is small relative to other measurement uncertainties. We therefore also propagate the errors associated with determination of

$u_{*th}$, random measurement error and imputation (gap-filling) error to annual estimates of net carbon exchange, and assess their magnitude relative to biases due to neglect of storage.

## 2 Methods

### 2.1 Site description

The site became operational in December 2011 (see Table 1 for site characteristics). It was established as part of a project investigating the concurrent effects of catchment reafforestation on biodiversity, carbon sequestration and stream water yields.

From an eddy covariance perspective, the site is considered 'ideal' in that it is relatively flat and homogeneous within the footprint area, and the canopy is very open (leaf area index ≈ 1). While the tower is situated on a slope of approximate south-easterly aspect (Figure 1), the slope is generally less than 1º. Vegetation is relatively homogeneous, consisting of a sparse eucalypt overstorey (dominant species is E. microcarpa, stand leaf area index [LAI] ≈ 1; see Table 1) and a sparse shrub understorey. The open canopy reduces the potential for strong decoupling of the sub-canopy space from the overlying air, and

underestimation of respiration by above-canopy eddy covariance systems relative to direct chamber measurements correspondingly decreases with declining leaf area (Speckman et al., 2014). Mean annual temperature for 2012-2014 was 15.9ºC (minimum and maximum temperatures of -3.1ºC and 45.0ºC). Average annual rainfall for the nearest long-term rainfall measurement site (Mangalore Airport; Bureau of Meteorology station ID 088109) is 560 mm (1971-2000 average).

### 2.2 Instrumentation

The eddy covariance (herein EC) method was used to measure $CO_2$ fluxes at 36 m (mean [±SD] canopy height was 15.3 [±6.4] m, but emergent individual trees were up to 30 m). The EC method requires fast-response instrumentation to measure simultaneous variations in scalar (here $CO_2$) and vector (3D wind velocities) quantities. A Campbell Scientific (Logan, USA) CSAT3 sonic anemometer (Table 2) was used to measure wind velocities and a Licor (Lincoln, USA) LI7500

infra-red gas analyser (herein IRGA) to measure $CO_2$ and $H_2O$ vapour mole fractions. EC data were logged at 10 Hz (post-processing is described below), and 30-minute averages for radiant and subsurface energy fluxes and standard meteorology (temperature, humidity, wind speed and direction, rainfall, barometric pressure) were also logged. All data were transferred telemetrically to a central server at Monash University, Clayton, Australia.

A custom-built profile system using a Licor LI840 IRGA measured changes in $CO_2$ storage below the EC measurement height. The system consisted of an array of 6 gas intakes (configured logarithmically in the vertical - 0.5, 2, 4, 8, 16, 36 m) connected to a sampling system (Figure 2) via tubing (of equal length – 18m – for all heights; enclosure mounted at 18 m). A KNF Neuberger (Freiberg, Germany) NMP 850.1.2 vacuum pump drew air from all levels through a common manifold with sample and exhaust chambers. A bank of 12V SMC (Tokyo, Japan) VO307 solenoid valves switched each of the gas lines

sequentially to a sampling loop (flow rate = 0.5lpm) consisting of a gas analyser (Licor LI840) and Alicat Scientific (Tucson, USA) mass flow meter, while the remaining lines bypassed the loop and were exhausted to the atmosphere. Dwell time for

each level was 20 s (first 15 s for flushing of the manifold, average of last 5 s logged), translating to a measurement cycle of 2 minutes.

Raw data retention was generally high over 2012-2014, with the station and all critical instruments running continuously,
except for 23/08/13 - 25/10/13, during which the profile system was damaged and had to be partially rebuilt. Data for this period are excluded from the analysis.

### 2.3 Data Processing and Analysis

Post-processing of EC data (including quality assurance and quality control) was undertaken using OzFluxQC, a software
package developed by the OzFlux community (primarily P. Isaac, the data manager of OzFlux) in the Python programming
language (Isaac et al., 2016, this issue). The Python programming language was used for all subsequent data analysis.
Thirty-minute fluxes were calculated from the 10 Hz data using block averaging (Moncrieff et al., 2004). Corrections applied
to the raw data included: 2-D coordinate rotation (Lee et al., 2004), in which the coordinate frame is rotated to force first the
mean cross-wind, then second the mean vertical wind, to zero over the measurement period; frequency attenuation
corrections, which account for loss of covariance associated with high frequency cut-off, sensor separation and path
averaging of then instrumentation (Massman and Clement, 2004), and; density corrections associated with the effects of
surface sensible and latent heat fluxes (Webb et al., 1980). Data were processed to level 4. In OzFluxQC, this represents the
point at which all QA / QC except $u_*$ filtering has been applied to flux data, and meteorological data have undergone QA /
QC and gap-filling (Isaac et al., 2016).

We identified the $u_*$ threshold ($u_{*th}$) using change-point detection (CPD) following Barr et al. (2013). The time series was
divided into multiple temperature-stratified samples, and a two-phase linear regression model was fitted to all possible
change points (change in $CO_2$ exchange as a function of $u_*$) for each sample. If the change point that minimises the sum of
squares error shows statistically significant improvement over a null model (no change point), the change point (i.e. $u_{*th}$) is
retained (subject to additional quality control criteria, as described by Barr et al., 2013). A bootstrapping procedure (in which
the data for each year were randomly sampled with replacement 1000 times; see Papale, 2006) was used to generate a
probability distribution for $u_{*th}$, the mean and 95% confidence interval of which provides a best estimate and uncertainty
interval for $u_{*th}$. Since we are assessing the effects of neglecting profile measurements in this study, we identified $u_*$
thresholds for the turbulent $CO_2$ flux alone and for the sum of turbulent $CO_2$ flux and $CO_2$ storage.

The rate of change of $CO_2$ storage was calculated from the difference between quasi-instantaneous (2-minute) vertical
concentration profiles at the tower at the beginning and end of the flux averaging period (Finnigan, 2006). We adopted the
approach for storage calculation of Yang et al. (2007):

$$\left(\frac{\Delta C}{\Delta t}\right)_{k=1} \times z_{k=1} + \sum_{k=2}^{n}\left\{\left[\left(\frac{\Delta C}{\Delta t}\right)_{k} + \left(\frac{\Delta C}{\Delta t}\right)_{k-1}\right] \times \frac{z_k - z_{k-1}}{2}\right\} \tag{1}$$

Here, $\Delta C / \Delta t$ is the time rate of change of $CO_2$ molar density (µmol $CO_2$ $m^{-3}$), $k$ is the profile level, $z$ is height above the surface (m) and $n$ is the number of profile levels. Mole fraction reported by the IRGA was converted to $CO_2$ molar density using the ideal gas law (temperature measurements were drawn from instruments co-located with the air inlet for each profile level, whereas pressure measurements were drawn from ground level). The 2-minute period preceding the beginning and the end of the 30-minute period (e.g. 1128-1130 and 1158-1200 for the 1130-1200 period) were used to calculate $\Delta C$. Given that the pump draw was simultaneously divided across 6 lines, there was a lag > 1 minute, so that the sampling was approximately temporally centred on the half hour. The average of the time derivative for two levels ($k$ and $k$-1) is the best estimate of the time derivative for the *layer* that has $k$ and $k$-1 as its upper and lower boundaries, except for the lower layer, for which it is assumed that $\Delta C / \Delta t$ for $k$ = 1 is representative for the layer. Layers were scaled according to the layer thickness and the storage term represented the sum over all layers.

## 2.4 Analytical framework

We assess the $CO_2$-carbon balance in the familiar context of a notional control volume with an orthogonal coordinate system (Finnigan et al., 2003), the mass balance of which (neglecting horizontal turbulent flux divergence) is:

$$NEE = \underbrace{\overline{w'c'}(h_m)}_{1} + \underbrace{\int_0^{h_m} \frac{\overline{\partial c(z)}}{\partial t} dz}_{2} + \underbrace{\int_0^{h_m}\left(\overline{u}(z)\frac{\partial \overline{c}(z)}{\partial x} + \overline{v}(z)\frac{\partial \overline{c}(z)}{\partial y}\right) dz}_{3} + \underbrace{\int_0^{h_m}\left(\overline{w}(z)\frac{\partial \overline{c}(z)}{\partial z}\right) dz}_{4} \tag{2}$$

Here, NEE (net ecosystem exchange of $CO_2$) is the true source term, term 1 is the turbulent flux across the upper horizontal plane of the control volume at instrument height $h_m$ ($w$ is the vertical velocity, and overbar and prime denote mean and quasi-instantaneous fluctuation from mean, respectively), term 2 is the storage term integrated over finite time period ($t$) and control volume depth ($z$), term 3 is the sum of the advection components in the horizontal dimensions ($x$ and $y$, with corresponding vectors $u$ and $v$) and term 4 the vertical advection. We adopted the standard micrometeorological convention in which NEE is positive (negative) when the net transfer of carbon is from ecosystem to atmosphere (atmosphere to ecosystem). For brevity, throughout we use the term 'carbon balance' to refer to long-term (annual) ecosystem / atmosphere exchange of $CO_2$, and do not quantify the much smaller exchanges of non-$CO_2$ gas species ($CH_4$ and volatile organic compounds) and potential net lateral aquatic transfer of dissolved organic and inorganic carbon. In the following text, equation 2 is simplified to:

$$NEE = F_c + S_c + A_c \qquad (3)$$

Here, $F_c$ and $S_c$ are the turbulent $CO_2$ flux and storage terms, and the vertical and horizontal advection terms have been
collapsed to a single advection term ($A_c$). During the day, when turbulence is well-developed, the turbulent flux ($F_c$) is generally the dominant term, but at night, the other terms may become dominant under weak mixing. Following the identification of $u_{*th}$, data were rejected where $u_* < u_{*th}$. There is some debate as to whether data filtering should be confined to the nocturnal period or whether it should also be applied across the full diel cycle (Papale, 2006). While the main emphasis in this study is on the nocturnal case, we assess the effects of inclusion of additional diurnal filtering on annual
carbon balances. While nocturnal advection was not measured, we took a similar approach to that of Clement et al. (2012), and inferred advection as the residual of the terms in equation 3. Nocturnally, NEE is equivalent to ecosystem respiration (herein ER). While ER is unknown when $u_* < u_{*th}$, it can be estimated ($\widehat{ER}$) using an empirical model, the parameters of which are optimised for periods in which the sum of turbulent flux and storage approximates ER (*i.e.* when $u_* > u_{*th}$). Equation 3 thus becomes:

$$\widehat{ER} - F_c - S_c = A_c \qquad (4)$$

### 2.5 Model selection and imputation

Because our approach implicitly assumes the model is a reliable estimator of the true respiratory source term, a robust model selection procedure is paramount to a reliable analysis. We used a simple set of empirical functions describing the response
of respiration to temperature and soil moisture. Optimisation of model parameters was undertaken using the robust non-linear least squares implementation in the Python Scipy package. The optimised functions were used to estimate ER for $u_* < u_{*th}$ (and for subsequent gap-filling). Akaike's Information Criterion (AIC) was used to determine the most appropriate model. AIC is a likelihood criterion that penalises the addition of model parameters, thereby discouraging the use of models of spurious complexity, and is formulated as:

$$AIC = -2\ln(L) + 2k \qquad (5)$$

Where $L$ is the likelihood of the fitted model. We tested two approaches to model fitting. In the first, we simply fitted parameters to the entire dataset, and in the second we fitted some parameters at annual time step and additional parameters at
arbitrary time steps (see below). The second approach arguably allows respiratory variation associated with neglected independent variables to be implicitly accounted for. We used AIC to objectively rank the performance of models using both approaches, as well as to determine the ideal time step in the second approach.

We tested linear, simple exponential and Arrhenius-type temperature response functions, in combination with sigmoid soil moisture response functions as described in Table 3. Linear functions have little theoretical justification as models for respiratory temperature response, and it has also been argued by Lloyd and Taylor (1994) that Arrhenius-type models have

5    more theoretical basis than simple exponential models. However, it is possible that underlying ecosystem respiratory dynamics may be sufficiently obscured by the random error inherent in eddy covariance data that more theoretically justifiable models may nonetheless not be justifiable in practice. Moreover, since respiration is the sum of aboveground (autotrophic) and belowground (autotrophic + heterotrophic) components subject to differing climatic conditions, we also tested two-compartment models, similar to the approach of Clement et al. (2012) and Swanson and Flanagan (2001).

The model with the best performance overall was a combined temperature (Lloyd and Taylor, herein L&T) and soil moisture response model of the form:

$$\widehat{ER} = \underbrace{R_{10}\, e^{E_o\left(\frac{1}{T_{ref}-T_0}-\frac{1}{T-T_0}\right)}}_{1} \times \underbrace{\frac{1}{1+e^{(\theta_1-\theta_2 VWC)}}}_{2} \qquad (6)$$

For the temperature response function (term 1), $T$ is the measured air temperature (see discussion below), $R_{10}$ is the respiration rate at the reference temperature ($T_{ref}$), $E_o$ is an activation energy parameter that controls temperature sensitivity, and $T_0$ is the temperature at which metabolic activity approaches zero. $T_0$ and $T_{ref}$ are fixed at –46.02 ºC and 10 ºC, respectively (Lloyd and Taylor 1994), since the unconstrained version of the function is overparameterised (Reichstein et al.,

20    2005; Richardson and Hollinger, 2005). The soil moisture response function (term 2; adapted from Richardson et al., 2007) is a sigmoid scalar response function that is constrained to the interval [0, 1] , and effectively modifies $R_{10}$ for the effects of low soil moisture. The function only accounts for effects of low soil moisture ($\theta_1$ and $\theta_2$ parameters control the x-intercept and gradient, respectively). We tested an alternative soil moisture response function (see Table 3) that accounted for effects of both low and high soil moisture, but performance was poor. This is most likely due to several factors: 1) the site is

25    primarily water-limited, such that there is limited data available to meaningfully model effects of high soil moisture; 2) the wettest period coincides with low soil temperature (winter) such that nocturnal signal: noise ratio for NEE is expected to be small, making it difficult to identify saturation effects, and; 3) the alternate 2-parameter soil moisture response function sacrifices flexibility at low soil moisture, which is likely to be the primary moisture-related constraint on respiration in this ecosystem.

The two-compartment L&T model also performed poorly, with and without soil moisture. This may reflect some combination of the low signal:noise ratio at this relatively low productivity site (see below) and / or the low soil carbon

storage at this site (Table 1). Low soil carbon does not *necessarily* imply that the belowground respiratory contribution must be small, since it may be driven by a combination of autotrophic (root) and heterotrophic (supported by exudates from the roots) respiration. However, we also tested numerous weighted combinations of soil and air temperature as inputs for Equation 6, and found that: i) a weighting of 0 for soil temperature and 1 for air temperature produced the lowest RMSE

(which will also translate to the lowest AIC, since there is no change to the number of parameters), for all air temperature heights tested (0.5, 2, 4, 8, 16, 36m); ii) the worst-performing temperature was soil temperature, and the best-performing was 36m air temperature.

This is well above the average canopy height (15.3m – see Table 1). This is unexpected since the temperature which yields

the best model prediction should be the temperature associated with the ecosystem stratum where the maximum respiratory production is occurring. However, the 36m temperature measurement shows very similar dynamics to the surface radiant temperature, both as a function of time of night and of $u_*$ (Figure 3). The surface radiant temperature in this case aggregates ground, trunk space and canopy level temperatures (since the canopy is very open – LAI $\approx$ 1.1), and when used as the temperature driver for model optimisation, produced an almost identical RMSE to 36m temperature (0.953 versus 0.951

$\mu molCO_2$ $m^{-2}$ $s^{-1}$, respectively), lower than for all other temperature measures. We hypothesise that a substantial proportion of $CO_2$ may be sourced from the litter layer, which contains over four times more carbon than the upper 40cm of the soil (Table 1). Litter contributions to soil respiration exceeding 50% have been reported in the literature (e.g. Xiao et al., 2014).

Other authors (e.g. Lasslop et al., 2010; Reichstein et al., 2005) have used the temperature response function in isolation,

fitting $E_o$ at annual timescale and allowing $R_{10}$ to vary on a smaller time step. Thus variations in $R_{10}$ may implicitly capture the effects of environmental controls (such as soil moisture) not explicitly included in the model. This was considered inappropriate for this ecosystem, since there may be rapid responses of ER to episodic rainfall following dry conditions at timescales shorter than the chosen step. Moreover, because soil moisture constraints coincide with high summer temperatures, this can force the $E_o$ parameter to extremely small values (i.e. such that the respiratory response is close to

zero) when the complete seasonal cycle is included in the parameterisation. We nonetheless found the best performing version of the above model (i.e. lowest AIC) used an annual time step for fitting of the $E_o$, a, and b parameters – simultaneous fitting of soil moisture response ensures that the $E_o$ parameter is not artificially reduced by effects of soil moisture. The remaining $R_{10}$ parameter was then fitted on a 7 day time step (with $R_{10}$ for intervening days linearly interpolated), which was chosen because it yielded the lowest AIC (Figure 4a; the only other candidate model with a

likelihood even close to – although nonetheless inferior to - this model was the exponential model combined with the same sigmoid soil moisture response function, fitted with the same time step). This suggests that there is real sub-annual variation remaining in the ER signal at frequencies of $\leq$7 days, which may include synoptically or seasonally driven effects. While additional real respiratory variance may also be present at frequencies >7 days, the results suggest that they cannot be separated from the noise.

While the coefficient of determination ($r^2$) for the chosen model was low (0.18), this appears to be primarily a function of the relatively low respiratory $CO_2$ production associated with this low-productivity site. Since there is irreducible random error in eddy covariance and profile measurements (Hollinger and Richardson, 2005), even a 'perfect' model cannot account for 100% of variance in observations. Since random error as a function of flux magnitude has a non-zero intercept and modest gradient (see section 3.5 and Figure 13), the relatively low nocturnal respiration at the site results in low signal:noise ratio nocturnal EC measurements at the site. This is compounded when NEE represents the sum of the turbulent and storage fluxes, because the latter substantially additionally increases random error (see Finnigan, 2006 for further discussion). Thus higher $r^2$ values observed in the literature (e.g. >0.6; Clement et al., 2012) most likely pertain to more productive ecosystems with a strong seasonal cycle. To test the effect of random error on model parameterisation, we used a procedure to characterise the random error in our measurements (see section 2.6 below), then superimposed this error on the pure model signal produced using the parameters derived from the optimisation in combination with the observational temperature and soil moisture data (Figure 4b). At higher temperatures (>22°C), the model tends to somewhat overestimate ER. We suspect this reflects the fact that these temperatures occur during a period of prolonged dry conditions, when the effects of low soil moisture could have sustained effects (e.g. episodes of drought-induced leaf senescence) that are not captured through an instantaneous soil moisture response. These instances represent <3% of all data.

The variance in the synthetic data explained by the model (13%) was actually smaller than for the observational data (18%). This apparent paradox is explained by the fact that the methodology for characterising random error is known to overestimate random error (by a factor of 1.5 to 2; Billesbach, 2011; Dragoni et al., 2007). Reducing the random error by these amounts results in $r^2$ of, respectively 0.25 and 0.37 (see Fig x). Thus a perfect model degraded with a conservative estimate of random error would still only explain 37% of the variance in NEE at maximum. Correspondingly, up to 20% (38 – 18%) of the unexplained variance in our observations may be associated with model error (due to missing drivers or mischaracterisation of the relationship between drivers and response).

We nonetheless ascertained that the $R_{10}$ parameter from the model could be reliably extracted at the chosen time step even with the inflated random error estimate. We ran $10^4$ trials, in each of which ER estimated by the model was degraded with a realisation of random error and the parameters then estimated again (Figure 4c). The 95% confidence interval shows that the dynamics of $R_{10}$ were reproduced even when the data were degraded (again suggesting that the variability in $R_{10}$ is real, rather than an artefact of random error). Estimated cumulative annual ER ± resulting uncertainty (95% confidence interval for the mean) was 995 ± 32 (3.2%), 905 ± 57 (6.3%) and 1028 ± 34g C m$^{-2}$ a$^{-1}$ (3.3%) for 2012, 2013 and 2014, respectively (the higher error in 2013 reflects missing data during late winter / mid spring, as previously noted). This approach implicitly assumes that the initial model is a true descriptor of reality, so we can conclude little about the potential role of the above-

noted model errors, but it does allow us to conclude that random error alone is not a barrier to reasonably robust parameter estimation, and that associated parameter and corresponding ER estimation errors are relatively small.

While the emphasis in this study is on the effects of nocturnal data treatment, gap-filling was also required for daytime to assess the effects of nocturnal data treatment on annual NEE. We used a Michaelis Menten-type rectangular hyperbolic model (Ruimy et al., 1995) of modified form (Falge et al., 2001) to estimate NEE, where ER was calculated from equation 5 using daytime temperatures in conjunction with nocturnally-derived parameter estimates:

$$NEE = \frac{\alpha Q}{1 - Q/2000 + \alpha Q/\beta} + ER. \tag{7}$$

Here, $\alpha$ is the initial slope of the photosynthetic light response, $Q$ is photosynthetic photon flux density, and $\beta$ is photosynthetic capacity at 2000 μmol photons m$^{-2}$ s$^{-1}$. The same step size and interpolation procedure was used as for the nocturnal fitting of the respiration model. We adopted the additional light-response model criterion in which $A_{opt}$ is modified to include a non-linear scaling factor to account for the effects of vapour pressure deficit (VPD) on stomatal conductance (Lasslop et al., 2010):

$$\beta = \begin{cases} \beta_0\, e^{(-k(VPD - VPD_0))}, & VPD > VPD_0 \\ \beta_0, & VPD < VPD_0 \end{cases}, \tag{8}$$

Here, $VPD_0$ is a threshold value above which stomatal conductance becomes sensitive to VPD, and $k$ is a fitted parameter defining the $\beta$ response to VPD.

**2.6 Uncertainty estimation**

We quantified sources of uncertainty in the data arising from random measurement and model error. We calculated random error from a daily differencing procedure (Hollinger and Richardson, 2005). When differences in critical drivers are sufficiently small (<35 Wm$^{-2}$ for insolation, <3 ºC for air temperature, and < 1m s$^{-1}$ for wind speed), differences between NEE data pairs separated by 24 hours were considered to represent random error. Since random error in EC data is heteroschedastic and follows the Laplacian distribution, we calculated the standard deviation of the error ($\sigma[\delta]$) for $j$ samples in $i$ u$_*$ bins as:

$$\sigma(\delta)_i = \sqrt{2} \, \frac{1}{n_i} \sum_{j=1}^{n_i} |\delta_{i,j} - \overline{\delta}_i|. \tag{9}$$

$\sigma[\delta]$ was regressed on the flux magnitude to derive a linear relationship that was in turn used to estimate $\sigma[\delta]$ for each datum. Monte Carlo-style simulation was used to translate these estimates to annual uncertainty, whereby for each of $10^4$

5    simulations, estimates of random error for all observational data were aggregated over one year. This yielded a normal distribution of uncertainties, the $2\sigma$ bounds of which were taken as the annual uncertainty.

With respect to model error, we followed Keith *et al.* (2009) in which, for day and night conditions, a sub-sample of $10^3$ records with observational data was randomly selected from the annual dataset. A proportion of the observational data in the

10    subsample equal to the observed proportion of data missing in the annual time series was then replaced with model estimates, and the summed difference between the complete observational and gap-filled subsamples was calculated and expressed as a proportion of the observational sum. This was repeated $10^4$ times, and the $2\sigma$ bounds of the proportional error was calculated, and then applied to the annual sum to produce an absolute annual model error.

15    To combine uncertainties, we assumed independence of the random and model estimates and sum in quadrature:

$$\varepsilon_{tot} = \sqrt{\varepsilon_r{}^2 + \varepsilon_m{}^2}, \tag{10}$$

Here, $\varepsilon_{tot}$, $\varepsilon_r$ and $\varepsilon_m$ are combined total, random and model uncertainty, respectively.

Other unquantified sources of error may also be present. Barr et al. (2013) argued that one of the key sources of uncertainty in annual NEE is the estimation of $u_{*th}$. We used their bootstrapping approach to derive robust confidence intervals for $u_{*th}$, and we included estimates of the effect on annual NEE of setting $u_{*th}$ to the upper and lower bounds of the 95% confidence interval for $u_{*th}$.

## 3 Results and Discussion

### 3.1 Contribution of mass balance components to nocturnal carbon dynamics

A nocturnal relationship was observed between friction velocity ($u_*$) and $F_c$ (Figure 5), with $F_c$ declining quasi-linearly below $u_{*th} = 0.42$ m s$^{-1}$ (identified using change point analysis of $F_c$) and approaching zero at zero turbulence. There was little seasonal
variation in $u_{*th}$ (data not shown), and the estimates and uncertainty bounds for all years were similar (Table 4). Given that the primary abiotic respiration controls, temperature and soil moisture, showed no relationship with $u_*$ except at $u_* < 0.08$ m s$^{-1}$ (which is linked to declines in temperature), we interpreted this as an increase in the non-turbulent terms of the mass balance.

The decline in $F_c$ with declining $u_*$ was accompanied by a corresponding increase in $S_c$ because as turbulence is
progressively suppressed, $CO_2$ is expected (excluding advective losses) to be increasingly stored below the measurement height. The strong sensitivity of $CO_2$ accumulation to $u_{*th}$ below the measurement height was observed in raw time series data under varying $u_*$ (Figure 6); the $CO_2$ mole fraction responded very sensitively to variations in $u_*$, and the effect of $u_*$ crossing $u_{*th}$ is evident.

However, $S_c$ was not the only important additional term in the nocturnal mass balance. A $u_*$ dependent decline in $F_c + S_c$ was also evident (Figure 7) below a threshold of $u_* = 0.32$ m s$^{-1}$ (identified using change point analysis of $F_c$), although this was inherently more uncertain due to the much higher random error in storage relative to the EC measurements (Table 4). There is no plausible explanation for $u_*$-dependent declines in the respiratory $CO_2$ source (again, excluding changes in relevant controls), and so we infer that this represents $CO_2$ losses associated with the remaining mass balance terms (*i.e.* advection).
This can be quantitatively estimated as a residual following equation 4. Note that parameter optimisation of the temperature response function used $F_c + S_c$ as the target variable because $S_c$ is observed to be non-zero when $u_* \gg u_{*th}$ (see Figure 5, and subsequent discussion).

The terms in equation 4 are plotted as a function of $u_*$ in Figure 7. The inferred advection estimate increased rapidly below $u_*$
threshold $= 0.32$ m s$^{-1}$, and was comparable to $S_c$ at the lowest $u_*$ values. This indicates that under the calmest conditions, $F_c$ approached zero, and approximately half of the $CO_2$ respired by the ecosystem was stored below the measurement height while the remainder was advected away. Integrated over the interval $0 < u_* < u_{*th}$, $S_c$ accounted for 61% of the difference between $F_c$ and $\widehat{ER}$, with the other 39% attributed to the advection components ($A_c$). This indicates that even on relatively flat terrain such as observed at this site, the nocturnal advection term is significant.

Since advection processes are not necessarily confined to the night, we also conduced the same analysis for the daytime. Daytime data were filtered using the nocturnally derived $u_*$ threshold derived for the sum of the turbulent flux and storage terms (0.32m s$^{-1}$). Over most of the range of u* values, on average there was little departure between the observations and

model (Figure 8), except at very low u∗ (and to a lesser degree at the highest values of u∗). Assuming that advection is responsible for model / data departures as a function of u∗, this indicates some possible residual early morning advection, since this is when the overwhelming majority of low u∗ daytime conditions occur.

## 3.2 Inferred advection mechanisms

$S_c$ for the individual layers is presented in Figure 9. There was a clear decline in $S_c$ for all layers below 8 m when u∗ was less than approximately 0.25 m s$^{-1}$. In contrast, storage in the higher 8-16 m and 16-36 m layers continued to increase near linearly. We hypothesise that these results indicate the onset of drainage flows at low levels under stable conditions, causing horizontal advective losses of $CO_2$ preferentially from the lower layers of the control volume. Since the presence of advection is inferred, it is not possible to directly assess the relative contributions of horizontal and vertical advection, but several factors indicate that horizontal advection is most likely the dominant mechanism.

Both the presence of a long and relatively consistent upward slope to the northwest of the tower (3-4km to the ridgeline - see Figure 1) and often clear and calm nocturnal conditions associated with the site's low altitude and continentality are conducive to the development of terrain-induced flows. This may be offset to some degree by the open, sparse canopy structure which, while aiding surface radiant heat loss, also inevitably reduces decoupling of sub-canopy meteorological conditions from the overlying atmosphere (Speckman et al., 2014).

The presence of katabatic flows alone does not imply advective $CO_2$ loss from the control volume - there must also be horizontal inhomogeneity in the $CO_2$ scalar field (see Equation 2). This may arise when the upwind flux source area (or footprint) extends beyond the limits of homogeneous vegetation cover. If the footprint at Whroo exceeds the distance to the local ridgeline under katabatic flow conditions, relatively $CO_2$-depleted air will be mixed downward and entrained into the surface flow. The midslope location of the site is also key to horizontal advective $CO_2$ loss; by contrast, for towers in valley bottoms, $CO_2$ would instead be expected to accumulate, likely resulting in advective $CO_2$ gain and corresponding *increases* in $S_c$.

Moreover, drainage flows are generally confined to the trunk space below the canopy (Aubinet et al., 2003). At the study site, mean canopy height was $15.3 \pm 6.4$ m (SD; Table 1). Conservatively assuming that the canopy comprises the upper 30% of tree height, drainage flows may be confined to depths of 10 m, comparable to commonly reported values (Goulden et al., 2006; Mahrt et al., 2001). The ongoing increases in storage in the 8-16 and 16-36 m layers suggests that the $CO_2$ source in these layers originated primarily from the vegetation rather than vertical transfer from lower layers.

In the interval between the $u_*$ thresholds for $F_c$ alone and $F_c + S_c$ (i.e. $0.32 \leq u_* \leq 0.42$ m s$^{-1}$), $A_c$ was not significantly different to zero (see Figure 7). The linear relationship between each of the lower layers (0-0.5, 0.5-2, 2-4 and 4-8 m) and the mean 8-36 m layer in the interval $0.32 \leq u_* \leq 0.42$ m s$^{-1}$ (i.e. the change-point derived $u^*$ threshold for $F_c + S_c$ and $F_c$, respectively) can be used to approximate the expected rate of change for those layers in the absence of advection when $u_* < 0.32$ m s$^{-1}$. Extrapolation of this linear relationship to conditions where $u_* < 0.32$ m s$^{-1}$ provides an estimate of the expected magnitude of $S_c$ in the absence of advection. If drainage flows are the primary advective mechanism, then the sum of $F_c$ and the linearly adjusted storage term should approximate $\widehat{ER}$. The correction to the storage in the 0-8 m layers when $u_* < 0.32$ m s$^{-1}$ increased the 0-36 m storage term such that for the interval $0 < u_* < 0.32$ m s$^{-1}$, the mass balance was approximately closed because $\acute{ER} - F_c \approx S_c$ to within the uncertainty (95% confidence interval) of the bin means over this interval (Figure 10). This indicates that the decline in $S_c$ at lower layers was of approximately the same magnitude as the inferred advection, consistent with the presence of low-level drainage flows removing $CO_2$ from the control volume.

The presence of katabatic flow may be observed as vertical shear in wind profiles, with terrain-aligned flow at low levels (below canopy), and synoptically aligned flow at higher levels (above canopy). Unfortunately, the installed wind instrumentation (RM Young Wind Sentry set – see Table 2) lacks the requisite resolution (minimum detectable wind speed = 0.5 m s$^{-1}$; minimum speed required to effect 10$^{\circ}$ deflection in wind direction = 0.8m s$^{-1}$) to detect weak drainage flows characteristic of moderate vegetated terrain (typically less than 0.5 m/s; Aubinet et al., 2003; Goulden et al., 2006; Mahrt et al., 2001).

While temperatures at lower levels also declined (slightly) more rapidly than at higher levels (Figure 3), this cannot plausibly explain low level storage declines. While cooler lower-level temperatures and lower $u_*$ are causally interrelated, they also have opposing effects on $S_c$: whereas decreasing temperatures reduce ER, decreasing $u_*$ increases the diversion of respired $CO_2$ into $S_c$ relative to $F_c$ (absent $A_c$). Thus the steady temperature decline relative to $u_*$ cannot explain the abrupt change in sign of the relationship between storage and $u_*$ at low levels.

### 3.3 Effects of correction methods on diel and annual NEE

The importance of the mass balance components to the diel carbon balance is presented in Figure 11 (note the data presented in the main figure – and discussed below, except where indicated – are based on the use of a diel $u_*$ filter). Given that the contribution of $S_c$ must average approximately zero over the diurnal cycle, annual NEE sums for both $F_c$ and $F_c + S_c$ were comparable: approximately –450, –400 and –560 gC m$^{-2}$ a$^{-1}$ for 2012, 2013 and 2014, respectively (Table 5; small differences [$< 20$ gC m$^{-2}$ a$^{-1}$] were observed for $F_c$ versus $F_c + S_c$, reflecting small differences in parameters of functions used for gap filling). However, the amplitude of the diel cycle also increased with the addition of $S_c$ (as well as a small daytime phase shift in peak $CO_2$ uptake: from midday – synchronous with the solar radiative peak – on average, to about 1100). The

application of the u∗-correction increased the nocturnal respiration estimate substantially, indicating the presence of advection under weak turbulence, as previously discussed. As previously noted, there was also evidence of continued advection following sunrise, which is expected given that substantial surface heating is required to reverse surface inversion conditions. Advection decreased to negligible levels after approximately 0800. Following u∗ filtering of these advective

effects, estimated annual carbon uptake was reduced by 60–70 gC m$^{-2}$ a$^{-1}$ (depending on year), resulting in a best estimate for annual NEE of -380, -326 and -486gC m$^{-2}$ a$^{-1}$ for 2012, 2013, and 2014, respectively.

The inset plot in Figure 11 shows the difference in NEE dynamics over the diel cycle arising from application of the 24 hour versus nocturnal-only u∗ filter. For $F_c + S_c$, the difference was small, and mostly confined to the early morning, when, as

noted above, the additional filtering most likely accounts for a small amount of residual advection between 0600-0800. The corresponding net decrease in annual C uptake was small (<±10g C m$^{-2}$ a$^{-1}$); however, it was also inconsistent between years, indicating the probably presence of more complex dynamics during this transition period.

Given that the majority of sites both internationally and in Australia do not have profile measurement systems, below we

discuss the effects of neglect of $S_c$ because this is the *de facto* approach taken for sites without profile systems. As a secondary option, a single point storage term (herein $S_{c\_pt}$) can be derived from the EC gas analyser. This will increasingly underestimate storage for taller towers, where much $CO_2$ accumulates within the control volume, and is subject to substantial error (Gu et al., 2012; Yang et al., 2007) but may nonetheless potentially reduce bias.

On average, $S_{c\_pt}$ performed very poorly nocturnally (relative to $S_c$), strongly underestimating $CO_2$ accumulation within the control volume (Figure 12). This is expected, since the profile system is required because the accumulation of $CO_2$ principally occurs below the measurement height in the absence of well-developed turbulence. Performance was also poor during the early morning transition, when the change in sign of $S_{c\_pt}$ lagged $S_c$ by up to two hours. However, when summed with $F_c$, and filtered for low u∗ conditions (u∗$_{th}$ for $F_c + S_{c\_pt}$ as determined by change point analysis was 0.33, 0.30 and

0.35m s$^{-1}$ for 2012, 2013 and 2014, respectively), performance was substantially improved, as long as the filtering was applied across the diel cycle. This corrected the nocturnal data, and excluded a substantial proportion of the problematic early morning data such that the diel dynamics were relatively similar to u∗-corrected $F_c + S_c$ (Figure 11). Compared with nocturnal-only u∗ filtering, diel u∗ filtering of $F_c + S_{c\_pt}$ slightly increased estimated $CO_2$ release before sunset, but substantially increased $CO_2$ uptake just after sunrise (inset plot in Figure 11). In fact, the peak in the increased uptake after

sunrise (-0.6 µmolCO$_2$ m$^{-2}$ s$^{-1}$) was similar in magnitude to the shortfall in storage from $S_{c\_pt}$ relative to $S_c$, thus bringing the early morning NEE dynamics for diel u∗-corrected $F_c + S_{c\_pt}$ into approximate agreement with the NEE estimate derived using $S_c$ from the profile system.

This correspondingly yielded estimates of relatively low bias ($\pm$ <20g C m$^{-2}$ a$^{-1}$) in the diel u$_*$-corrected estimates relative to $F_c + S_c$ for years 2012 and 2014, but caused more substantial overestimation of the net carbon sink for 2013. For both 2012 and 2014, the reported annual C sink was increased, because much of the data during the early morning when $S_{c\_pt}$ tended to be too high was removed, and replaced with data modelled using higher u$_*$ conditions (under which $S_{c\_pt}$ was a smaller

component of the mass balance relative to $F_c$, thus reducing bias). The inconsistency in 2013 arose due to the combined effects of error on model parameterisation and subsequent imputation, in combination with the effects of missing data. Random error estimates (derived from equation 8) are shown in Figure 13; at low flux magnitudes the error associated with $F_c + S_{c\_pt}$ was large relative to that for $F_c$ alone or $F_c + S_c$. This may be due to a mismatch in the timing of peak $S_{c\_pt}$ relative to $S_c$; small day-to-day variations in this timing may – in conjunction with noise in $S_{c\_pt}$ – result in large variations in NEE

estimates for a given set of environmental conditions (as essentially demonstrated in Figure 13). Large errors in point-based storage estimates were also reported by Gu et al. (2012) for a forest ecosystem. This in turn affects the parameter estimates for $R_{10}$ in the respiration function (equation 5) and $\alpha$ in the light response function (equation 6) in particular. There was a prolonged period of missing data during 2013 (late August to late October), and the estimate of NEE during that period is strongly dependent on the parameter estimates immediately preceeding and succeeding the data gap. When these estimates

are subject to greater error, the interpolated values in the gap are similarly subject to greater error. This effect can also be seen in the larger uncertainty bounds for $R_{10}$ estimates derived from u$_*$-filtered $F_c + S_c$ presented in Figure 4a.

Correction of $F_c$ alone (using diel u$_*$ filtering) yielded comparable nocturnal NEE to that estimated from u$_*$-corrected $F_c + S_c$ because it implicitly accounted for the nocturnal effects of both storage and advection by eliminating data where those terms

were large (Figure 11). There was still a small shortfall nocturnally, because, particularly in the early evening, $S_c$ was on average >>0 when u$_*$ > u$_{*th}$, such that the respiration model used to fill the gaps was unavoidably optimised with a biased estimate of $F_c$. The NEE estimate during the early morning was improved relative to uncorrected $F_c$, again because this removed data during the period when $S_c$ was relatively large and $F_c$ a correspondingly biased estimator of NEE. The effect of filtering low u$_*$ conditions during the day can also be seen in the inset panel of Figure 11, and had a similar effect to daytime

filtering of $S_c + F_{c\_pt}$. However, as noted with the nocturnal case, in the early morning, the magnitude of $S_c$ was initially large even after u$_*$ > u$_{*th}$. As a result, the $F_c$ data used for optimisation were still biased.

Why was $S_c$ non-zero when u$_*$ > u$_{*th}$? In theory, in the absence of a net $CO_2$ source / sink or advective $CO_2$ loss, the presence of a vertical gradient in $CO_2$ density at the morning resumption of turbulence must result in turbulent efflux of $CO_2$ from the control volume, which in turn must – as dictated by mass conservation – be balanced by a change in storage of equal and

opposite magnitude. Under such conditions this would be expected to manifest as a spike in $F_c$ (for example, see Aubinet et al., 2012, and Figure 5.2 therein).  However, the net source / sink term is rarely zero. In the morning, photosynthetic activity began at approximately 0600 on average, as evidenced by the sudden decline in $S_c$ at this time (Figure 12), crossing zero at approximately the time that u$_*$ reached its diel minimum. The ecosystem-level light compensation point (when GPP $\approx$ ER, as

indicated by $F_c + S_c = 0$) occurred between 0700-0800, with net photosynthetic $CO_2$ uptake thereafter (see Figure 11). By consuming $CO_2$ within the control volume, photosynthetic activity necessarily reduces the vertical $CO_2$ gradient and thus commensurately reduces the turbulent flux across the upper plane of the control volume (the situation is more complex than this: when turbulence is weak, storage changes within the canopy may not have a substantial effect on the $CO_2$ gradient

across the upper plane of the control volume, such that $F_c$ and $S_c$ are not immediately coupled). As such, there was no identifiable spike in $F_c$. But whether the change in storage occurs as a result of turbulent ventilation of $CO_2$ from the control volume or photosynthetic uptake, this change is missed by the EC measurement system and NEE estimates based on $F_c$ alone are consequently biased high. Over time, turbulent transfer and photosynthetic consumption steadily eliminates the accumulated $CO_2$ surplus in the control volume, but the process is not instantaneous, thus the presence of large initial early

morning $S_c$ even when $u_* > u_{*th}$.

Early morning photosynthesis may be particularly pronounced in eucalypt-dominated ecosystems because the characteristically pendulous (in some species up to 75% of mature leaves typically hang at angles > 80 degrees from horizontal; Pereira et al., 1987), amphistomatous leaves evolved to maximise incident radiation at low sun angles. This shifts

photosynthetic activity towards periods with lower vapour pressure deficit, reducing water losses (James and Bell, 1996). Mutual shading would partially counteract this effect at low sun angles, but this may have less effect in systems with sparse canopies such as the woodland in this study. An Australian temperate eucalypt forest site with long-term turbulent flux and $CO_2$ profile measurements also showed no morning spike in $CO_2$ efflux (Van Gorsel et al., 2007, Figure 4 therein).

Effectively the reverse process began before sundown. The net surface $CO_2$ sink transitioned to a source between 1700-1800, as GPP declined relative to ER (because temperature, the key driver of respiration, lags insolation, the key driver of photosynthesis, while higher vapour pressure deficit may additionally induce partial stomatal closure). This was accompanied by a rise in the quantity of $CO_2$ partitioned into $S_c$ despite $u_* > u_{*th}$. This is most likely again related to the effects of the prior conditions on the vertical $CO_2$ gradient. The photosynthetic drawdown of $CO_2$ during the day resulted in a

$CO_2$ deficit within the control volume (relative to air at higher levels) that persisted until the evening. The minimum $CO_2$ mole fraction occurred at the afternoon zero crossing of $S_c$, which is simply the time rate of change of $CO_2$ mole fraction, scaled appropriately. Thus the change in sign of storage may result partially from the reversed concentration gradient (driving mixing of $CO_2$ downward from aloft), but by the late afternoon the system was an increasing net $CO_2$ source, such that a large proportion of the $CO_2$ entering the control volume from the ecosystem acts to increase the $CO_2$ mole fraction

within the control volume. Again, this is inevitably missed by the EC system, and thus NEE estimates based on $F_c$ alone are necessarily biased high. Over time, turbulent transfer and photosynthetic consumption steadily eliminates the accumulated $CO_2$ deficit in the control volume, but the process is not instantaneous, thus the presence of large initial early evening $S_c$ even when $u_* > u_{*th}$.

Note however that this early evening bias does not affect annual NEE estimates, because it is balanced against equivalent daytime biases. The diel dynamics of storage can be divided into two 12 hour periods, over each of which the sum of $S_c \approx 0$: 2100–0900 and 0900-2100 (these times are chosen because for 2100-0900, $CO_2$ mole fraction on average was greater than its mean value, whereas for 0900-2100, it was less). In the first period, if only $F_c$ is used to estimate NEE, a potential double-counting problem arises in the morning because of the fact that the neglected change in storage partially reflects the removal of nocturnally accumulated respired carbon which has already been accounted for by the nocturnal correction. In the second period, the underestimation of true NEE due to neglect of $S_c$ is balanced across the period (the sign and magnitude of $S_c$ between 0900 and approximately 1600 balances the period between 1600-2100).

Thus the corresponding annual NEE estimates derived from diel $u_*$-filtered $F_c$ were in all years biased towards net carbon efflux (by 47, 39 and 45 gC m$^{-2}$ a$^{-1}$ for 2012, 2013 and 2014, respectively) relative to the corresponding best estimate of NEE (i.e. diel $u_*$ corrected $F_c + S_c$). In comparison to uncorrected estimates of annual NEE (*i.e.* gap-filled raw data), this reduced the absolute bias, but reversed the sign. The performance was more consistent but on average more biased than for $F_c + S_{c\_pt}$. The higher bias is expected, because although using the diel $u_*$ filter resulted in comparable improvement in NEE across the diel cycle (see inset of Figure 11) for both $F_c$ and $F_c + S_{c\_pt}$, the improvement in $F_c$ was not added to an existing (albeit imperfect) estimate of $S_c$ as was the case with $S_{c\_pt}$. Thus some proportion of the contribution of the morning negative storage term to the mass balance was still missed, resulting in underestimation of $CO_2$ uptake on average.

Far more problematic for estimation of annual NEE was the effect of applying a nocturnal $u_*$ filter only. This unmasked the entire morning period (when $u_* < u_{*th}$) during which the magnitude of $S_c$ was relatively large and $F_c$ correspondingly diminished. The effect was such that applying the correction changed the sign of the bias (as in the above case), but also (on average) *increased* its magnitude; whereas the uncorrected NEE estimates were biased towards uptake by 66, 61 and 65 gC m$^{-2}$ a$^{-1}$, nocturnal $u_*$-corrected $F_c$ was biased towards efflux by 79, 60 and 92gC m$^{-2}$ a$^{-1}$. In other words, in the absence of storage estimates, a more accurate estimate of annual NEE would be obtained at this site if the nocturnal $u_*$-correction were *not* applied.

This is expected given the fact that $S_c$ contributed more to the nocturnal underestimation of $F_c$ than advection. Whereas advected $CO_2$ is (in this case) lost to the sensing system, the stored $CO_2$ causes an equivalent offset in the turbulent flux measurements when it is evacuated from the control volume in the morning (via the mechanisms described above), thereby biasing the system towards efflux. The mechanism is as described above, but is more severe when $u_*$ filtering is not extended to the day. If the proportional contributions of $S_c$ and $A_c$ are approximately equivalent, the application of nocturnal $u_*$ filtering to $F_c$ alone would be expected to preserve the magnitude but reverse the sign of the induced bias. In theory, where $S_c$ is dominant (in this study we ascertained a contribution of approximately 61%), the bias will necessarily be worsened by this treatment (since the quantity which needs to be corrected – $A_c$ – is smaller than the quantity which should *not* be

corrected but is implicitly - $S_c$). By the same logic, where $A_c$ is dominant, the bias should be reduced. In practice, sites with severe advection problems may also be subject to flow decoupling that renders u∗ filtering ineffective (e.g. Van Gorsel et al., 2007), but the issue is not confined to the u∗ filtering methodology *per se*. Any method of correcting nocturnal data that does not take account of the subsequent daytime effects imposed by the nocturnally stored $CO_2$ will be subject to the same issues.

### 3.4 Effects of neglecting $CO_2$ storage on physiological interpretation of data

From the perspective of deriving annual NEE sums, it is daytime rather than nocturnal measurements that are more critical; applying u∗ filtering to either $F_c$ or $F_c + S_c$ resulted in similar estimates of nocturnal NEE on average. But the effects of neglecting $S_c$ depend on time of night. $F_c$ underestimated NEE following sunset, even where u∗ > u∗th for $F_c$.

A secondary nocturnal problem is also recognised in the literature: when u∗ increases following extended calm periods, stored $CO_2$ is vented from the control volume, which artificially inflates $F_c$ relative to the true source term, the extent of which effect will depend on the importance of advection (Aubinet et al., 2012). When a u∗ threshold is imposed, such periods are likely to be included in the retained data. This has the opposite effect to the early-evening effect, and is more likely to be

15    problematic later in the evening when stable stratification and substantial storage of respired $CO_2$ is more likely. Both effects were observed in the nocturnal progression of $S_c$ for periods when u∗ > u∗th (Figure 14): $S_c$ was on average > 0 for the first 4-6 hours after sunset and < 0 afterwards. On balance, the effect in this study was to increase slightly the estimation of ER. This explains why $S_c$ was slightly positive when u∗ > u∗th in Figure 5, and why $F_c$ was slightly lower than $F_c + S_c$ at night (primarily in the early evening) in Figure 11.

However, given that temperature decreases over the evening, this suggests that the slope of temperature response functions will be slightly increased for $F_c + S_c$ versus $F_c$ alone. Given that the optimisation procedure minimises the prediction error, this may not have a large quantitative effect averaged over the evening, but interpretation of system response to temperature is distorted. Moreover, extrapolation beyond the parameterisation domain (e.g. estimation of daytime ER) may result in

25    substantial error because distortion of function parameters (e.g. $E_o$ and $R_{10}$ in equation 5) will potentially result in systematic error (because the function optimised using $F_c$ will underestimate NEE at high temperatures). Any systematic error in estimated daytime ER will then necessarily propagate to estimation of GPP (commonly calculated as NEE – ER). Because these errors are offsetting, this is not likely to have a large effect on annual NEE estimates.

30    Similar distortion of response to insolation occurs during the day. The addition of $S_c$ substantially affects diurnal NEE dynamics, particularly during the morning, which affects the interpretation of the controls on NEE. For example, Figure 15 shows the difference in radiation use efficiency (RUE – here simply defined as the ratio of mean NEE to mean insolation) during daylight hours for $F_c$ alone versus $F_c + S_c$. RUE was higher in the early morning, and declined more sharply, when

NEE $= F_c + S_c$. Such declines in RUE are often associated with stomatal response to increasing VPD, and so the importance of this driver may be missed or minimised when $S_c$ is not measured. Filtering low $u_*$ conditions during the day substantially improved the light response of $F_c$, again because it removed a large proportion of the data during periods when $S_c$ was large. It also moderately increased the RUE of $F_c + S_c$ in the early morning, again most likely reflecting the removal of the small previously identified effect of residual advection in the early morning when $u_* < u_{*th}$.

Application of light-response function analysis to daytime data to extract either photosynthetic or respiratory parameters is problematic in the absence of storage measurements because $F_c \neq$ NEE during most of the day. The estimation of ER (and quantum efficiency) derived from light response function analysis (e.g. Gilmanov et al., 2003; Lasslop et al., 2010) is strongly dependent on the magnitude of observed NEE when insolation is low (sunrise and sunset), and thus the effect of neglecting the storage term may be particularly distorting to these parameters (Aubinet et al., 2012).

### 3.5 Sources of uncertainty

While bias is to be avoided where possible, if the known effect of bias is small relative to the uncertainty in NEE, then it may not be of particular concern. Here we analyse the magnitude of several uncertainty sources relative to the magnitude of bias (given the inconsistency problems with the point-based storage estimate, here we only compare $F_c$ with $F_c + S_c$). One of the largest sources of uncertainty in annual NEE estimates is expected to derive from uncertainty in $u_{*th}$ (Barr et al., 2013; Papale, 2006). We propagated this uncertainty to annual NEE (for both $F_c$ and $F_c + S_c$) by filtering and gap-filling the data using the lower and upper bounds of the 95% confidence interval (CI) of the normally distributed population ($N = 10^3$) of $u_{*th}$ derived from CPD (Table 4). Much larger effects were evident for the lower uncertainty bound ($\mu - 2\sigma$, where $\mu$ is the best estimate for $u_{*th}$ and $\sigma$ is the standard deviation), which is to be expected because systematic errors in nocturnal flux measurement occur at low $u_*$. However, there should be no systematic variation in NEE when $u_* > u_{*th}$. The direct effect of the upper uncertainty bound ($\mu + 2\sigma$) is expected to be minimal. While the reduction of nocturnal data availability for higher $u_{*th}$ is expected to increase parameter imputation uncertainty, the effect here was minor, with annual NEE for $u_{*th} = \mu$ and $u_{*th} = \mu + 2\sigma$ differing by <10 gC m$^{-2}$ a$^{-1}$ in all years.

The uncertainty in $u_{*th}$ was greater for $F_c + S_c$ than for $F_c$ alone due to the additional random error inherent in $S_c$ (see Finnigan, 2006 for further discussion), which feeds into the change-point detection process. This propagated to larger uncertainty in the lower bound for annual NEE (50-75 gC m$^{-2}$ a$^{-1}$ compared to 20-40 gC m$^{-2}$ a$^{-1}$ for $F_c$). This is because for $F_c + S_c$, the lower bound of the $u_{*th}$ uncertainty is below the 1st percentile of the nocturnal data, such that the full effect of advection was propagated to annual NEE but for $F_c$ alone, it was closer to the 40th percentile, such that only a small proportion of storage and advection occurred in the interval between $u_{*th} = \mu$ and $u_{*th} = \mu - 2\sigma$. The best estimates of NEE for

$F_c$ alone versus $F_c + S_c$ are sufficiently different that their respective uncertainty ranges do not overlap, indicating that biases are large relative to u∗-induced uncertainty.

The large lower-bound uncertainty in annual NEE derived from $F_c + S_c$ is very likely overestimated. If the lack of a relationship between u∗ and $F_c$ above u∗th indicates that additional terms in the mass balance are negligible, then $S_c$ should on average approach zero at u∗th. Given that this is what we observed (Figure 5), and the measurement system (and associated measurement errors) for $S_c$ is independent of that for $F_c$, this is an independent validation of u∗th. It is not clear how such information might be used in the context of frequentist statistical analysis, but it strongly suggests that the uncertainty bounds for NEE that include the effects of u∗th uncertainty presented here are unrealistically large.

The NEE uncertainty contribution of combined random and model error was small ($\leq 30$ gC m$^{-2}$ a$^{-1}$) by comparison (Table 6), < 10% of annual NEE for each year. Model-induced uncertainty was generally larger than random uncertainty, but the difference was more pronounced at night. This is because far more nocturnal data was removed by u∗ filtering than daytime data. The removal of observational data correspondingly increases the model error (because more model values are used) and reduces random error (because random error is only compounded across the cases for which there are observations). In contrast, during the day, there is less model data, and random error on individual data points is generally larger because of larger fluxes in combination with the heteroschedasticity of random error (see Figure 13).

Annual NEE uncertainty due to random and model error was slightly greater for $F_c + S_c$ than for $F_c$ alone, due to additional random error to that for $F_c$ arising from the addition of $S_c$ (Figure 13). This was largely nocturnally determined because the storage term was smaller and less variable during the daytime when fluxes were largest. The increased annual uncertainty of $F_c + S_c$ was largely due to higher model uncertainty, which most likely reflects the propagation of random error to model uncertainty through its effects on non-linear parameter estimation. This was most pronounced nocturnally because of the lower signal to noise ratio of $F_c + S_c$ relative to the daytime.

The method of calculating model-induced uncertainty may overestimate model error. It compounds observation – model data differences, but the observational data already contain random error so this necessarily inflates the propagated model uncertainty above errors associated with missing driver information or systematic measurement error. This problem is then partially propagated to the daytime because random error contributes to uncertainty in the nocturnally derived parameters of equation 5, which are then used to calculate the ER component of daytime NEE (equation 6). On the other hand, since the model – data comparison inevitably only occurs for periods where observation are available, it cannot be ascertained whether model and observational data would depart more substantially under conditions where the observational data is missing (which are often more extreme than the conditions for which observational data are available). The total error may thus be somewhat constrained.

The interdependence of model and random error also technically renders invalid the assumption of independence in equation 9. However, the effect is to increase rather than to decrease uncertainty, which as noted above was nonetheless small. While there are methods to separate model and random components (see Dragoni et al., 2007), this generally requires co-location of two instrument arrays. However, the daily differencing procedure we used is known to overestimate error by up to a factor of 2 (Billesbach, 2011; Dragoni et al., 2007) due both to potential wind-dependent temporal variations in source / sink strength (which may materially affect annual NEE estimates; Griebel et al., 2016) and because some signal is captured in the differencing procedure.

The random error in $S_c$ may potentially be reduced by redesign of the intake ports of the profile system. At present, air is drawn through a single port (via a filter) into the sampling tubes. This is similar to the design of other commercially available systems. Multiple spatially distributed intake ports may act to smooth the effects of random eddies, although it is imperative that this does not result in a large increase in the effective averaging time for the system, which, as noted by Finnigan (2006), may reduce the magnitude of the calculated storage term.

It should be emphasised that there are numerous sources of uncertainty that have not been quantified here. Perhaps most important of these is systematic errors in the measurements themselves, which may be an extremely important source of true uncertainty (Lasslop et al., 2008). Thus the uncertainties reported here for $F_c + S_c$ also should not be formally interpreted as total uncertainty in the true source / sink term, but as the uncertainty contributed by a subset of quantifiable errors.

## 4 Conclusions

We used a simple method to infer nocturnal advection from measurements combined with a simple and widely used empirical temperature response respiration model (modified for the effects of low soil moisture). While the nocturnal signal to noise ratio was low in this relatively unproductive ecosystem, our results suggest that relatively accurate parameter

estimation - and corresponding estimation of ER - is still possible. Even at our very flat site, approximately 40% of nocturnal $CO_2$ flux underestimation was attributable to advection. Observation of reductions in storage at lower levels (within 8m of the surface) in response to declining $u_*$ indicate that the most likely advective mechanism is terrain drainage flows. High resolution measurements of wind directions within the control volume would be invaluable for directly detecting the presence of these flows, and are planned for this site.

For the ideal case of observational NEE measurements consisting of the sum of turbulent flux and profile-measured storage, we found that correcting for nocturnal respiratory underestimation (due to advection) using $u_*$ filtering reduced the cumulative annual $CO_2$-carbon sink estimate by 10-20%. Applying the $u_*$ filter across the diel cycle rather than nocturnally for this series had a small effect on cumulative annual NEE, very slightly reducing carbon uptake on average (though not

consistently for all years). The bias occurred in the early morning, and likely indicates residual drainage-induced horizontal advection after sunrise.

In the absence of storage estimates from a profile system, single-point estimated storage at this site generally biased cumulative annual NEE estimates towards efflux. Nocturnal $u_*$ filtering removed much of the erroneous nocturnal storage

data, such that on average nocturnal estimates were not substantially biased. However, much large biases (towards efflux) occurred annually when the $u_*$ filtering was only applied nocturnally. Application of the diel $u_*$ filter reduced bias substantially. However, the single point storage estimate greatly increased the error in measurements, and is likely to result in large uncertainties where significant data gaps are present.

The use of $F_c$ alone to estimate NEE also resulted in consistent bias towards efflux, but the bias was on average larger although more consistent relative to the use of the single point estimate. As above, the bias was far worse when $u_*$ filtering was only applied nocturnally. In fact, in this case the bias was larger than if the correction was not applied at all. This is likely to be the case for any site where the quantity of $CO_2$ diverted into storage over the night exceeds the quantity diverted into advection.

But this underscores the intractable nature of the problem: the relative contributions of storage and advection to the nocturnal mass balance cannot be quantitatively assessed in the absence of profile measurements. Moreover, even at sites where drainage flows are known to regularly occur at night, it is likely that shear-induced turbulence penetrates below canopy only

under strong winds; the rarity of such conditions may result in the rejection of an unacceptably large number of data. Where this is the case, profile measurements are required to increase the proportion of available nocturnal data because storage increases prior to the onset of drainage flows, which only occur once the cooling air mass adjacent to the surface achieves sufficient density to overcome friction and begin to flow (Van Gorsel et al., 2007).

Storage measurements nonetheless introduce some minor complications for data interpretation. The additional random error in nocturnal storage measurements increases uncertainty in $u_*$ threshold and, correspondingly, annual NEE (uncertainties due to direct random observation error and imputation error were small by comparison). But as we have argued, the lower bound uncertainty for $u_*$ threshold is unrealistic, since the storage term on average approaches zero at the $u_*$ threshold. This

10 behaviour is expected if the central $u_*$ threshold estimate from change point detection is approximately correct. Even if these uncertainties are considered accurate, when propagated to annual NEE, the resulting uncertainty intervals for the sum of turbulent flux and storage versus turbulent flux alone do not overlap. This indicates that biases are not subsumed within (quantified) uncertainties; effectively, profile measurements (slightly) reduce precision and increase accuracy of annual NEE estimates.

We therefore believe that for both OzFlux and Fluxnet, the installation of profile systems for sites with tall canopies (woodlands, forests, savannas) is extremely important to ensure that both determination of annual carbon exchange and interpretation of ecosystem processes are accurate. At the very least, the issues explored here need to be taken into consideration during data analysis, and alternative methods of estimating uncertainties at sites without profile systems need

20 to be developed. For sites under the auspices of the Integrated Carbon Observation System (ICOS: www.icos-ri.eu), profile systems are mandatory; while this is not yet the case for OzFlux and Fluxnet, for accurate estimates of annual NEE, profile systems are vital.

25 *Acknowledgements.* This work was made possible by funding from the Australian Research Council (ARC; Linkage Project 'More bang for your carbon buck: carbon, biodiversity and water balance consequences of whole-catchment carbon farming' [LP0990038]). IM would also like to thank Peter Isaac for his invaluable comments during development of the manuscript. Finally, IDM, JB, PJB, TRC, RM and RMT deeply regret the passing in September 2016 of SCC, who made significant contributions to this work.

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

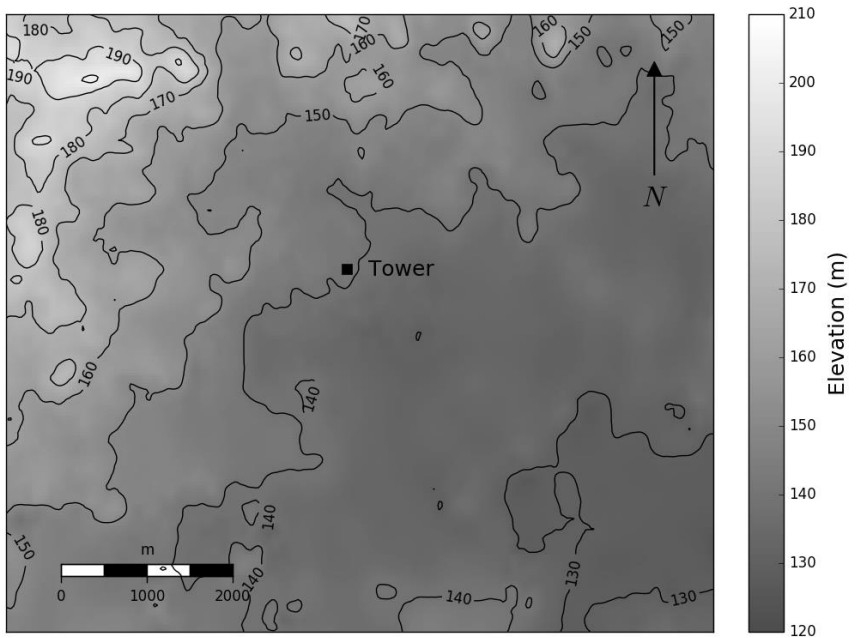

**Figure 1: topography of terrain surrounding tower.**

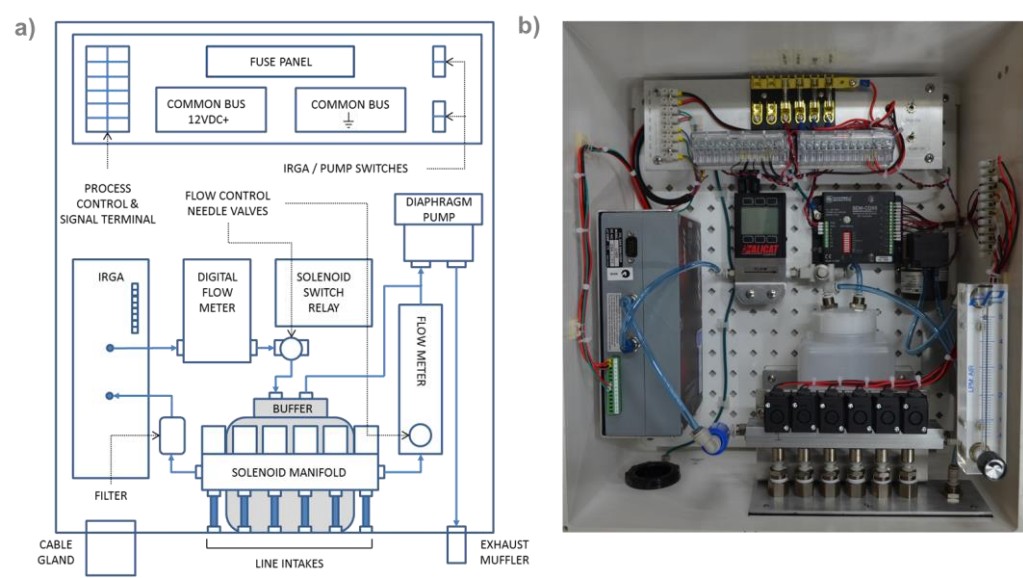

**Figure 2: a) schematic, and; b) photographic layout of profile gas analysis system.**

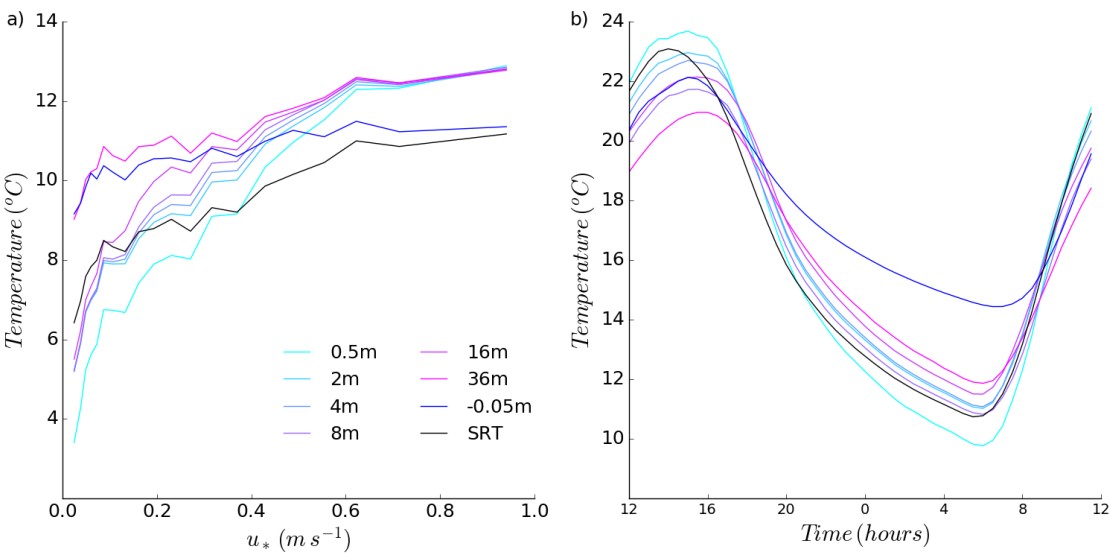

**Figure 3: dependence of temperature on: a) ustar, and; b) time of day (temperature measurement for -0.05m represents soil temperature, and SRT is surface radiant temperature).**

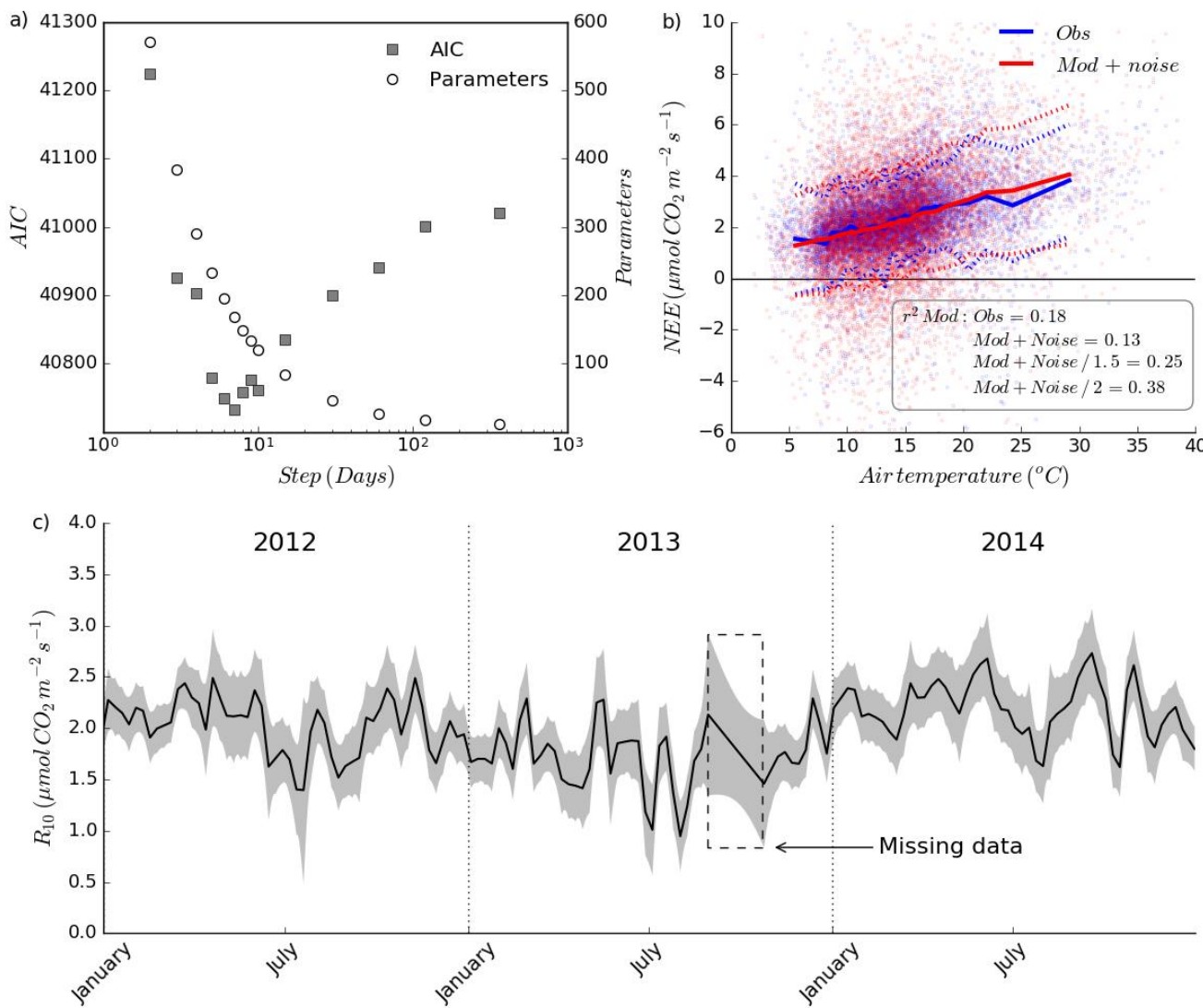

Figure 4: a) Akaike's Information Criterion (AIC; LH axis) and number of parameters (RH axis) for model with time step-varying $R_{10}$ component; b) comparison of temperature response of observational data with modelled data degraded with data-derived random error (dashed lines represent ±1 SD; observational points are shown for reference); c) interpolated $R_{10}$ estimates derived from observational data optimisation using seven day time step (black line), and 95% confidence interval for the mean (grey shading) for $R_{10}$ values extracted from $10^4$ trials in which ER predicted from the empirical model was degraded with data-derived random error.

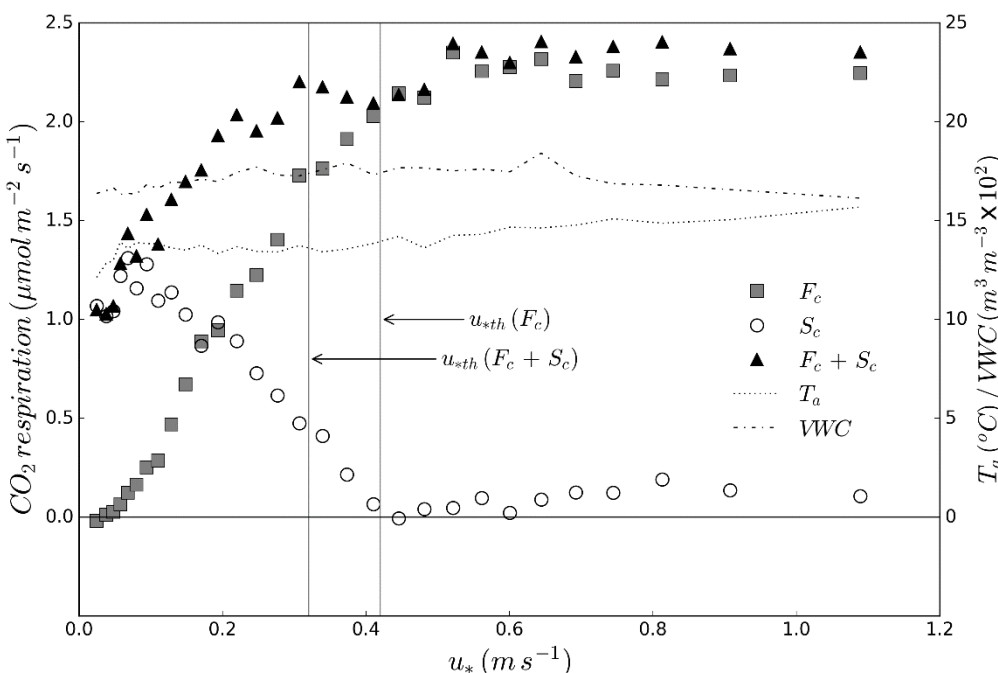

**Figure 5: LH axis) dependence of mean measured nocturnal carbon mass balance components (turbulent flux [F$_c$], storage [S$_c$] and F$_c$ + S$_c$) on friction velocity (u$_*$); RH axis) air temperature at EC instrumentation height (36m) and volumetric soil moisture content at 10cm. Vertical lines denote u*th for both F$_c$ and F$_c$ + S$_c$, as labelled.**

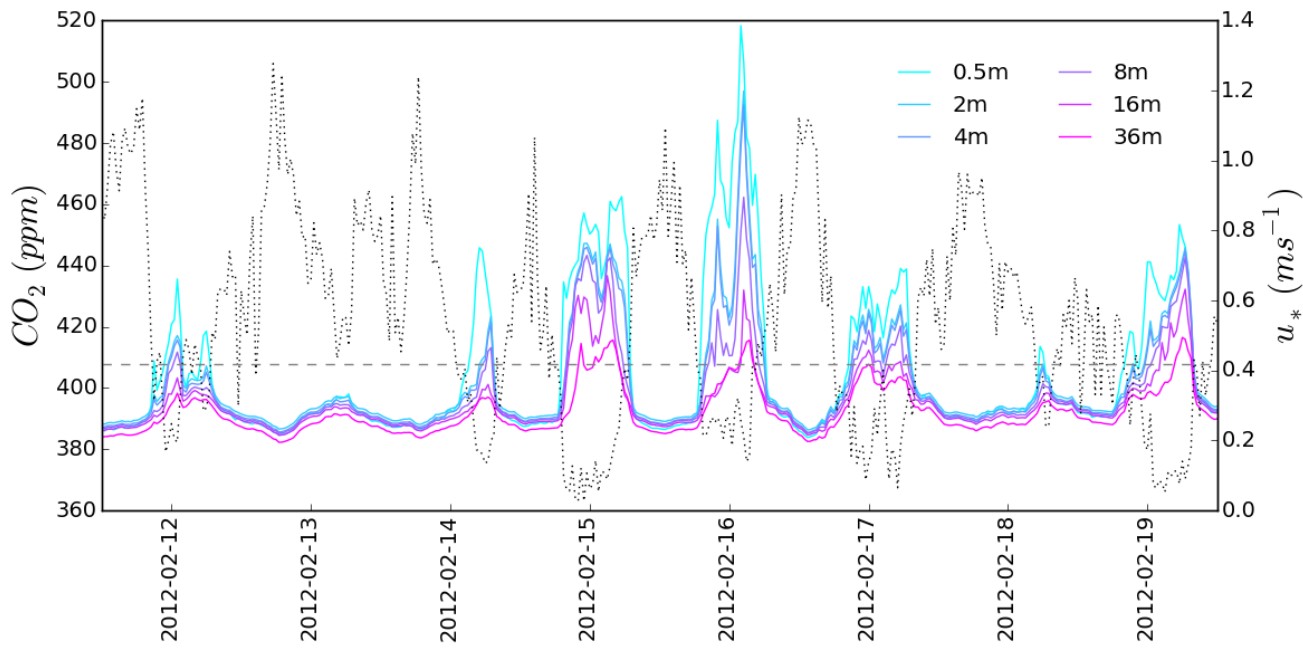

Figure 6: LH axis) profile system time series (date labels correspond to midnight) of $CO_2$ mole fraction (coloured solid lines); RH axis) time series of $u_*$ measured at 36m (grey dotted line). Note horizontal dashed line marks $u_{*th}$ for $F_c$.

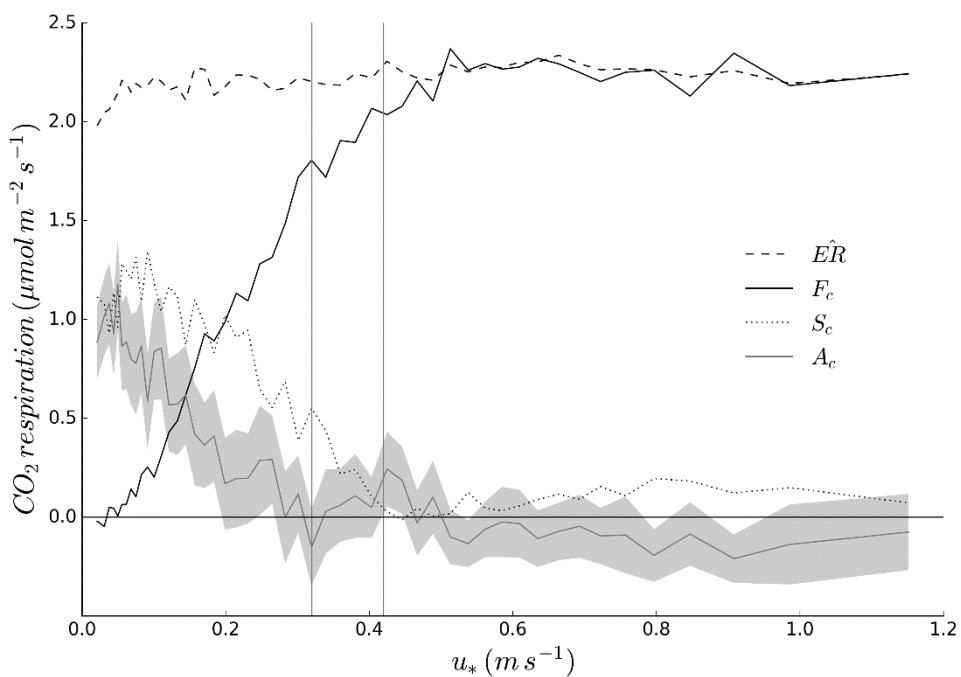

**Figure 7: dependence of measured ($F_c$, $S_c$), model-estimated ($\widehat{ER}$) and inferred ($A_c$) mass balance components on friction velocity ($u_*$; the grey shaded area represents the 95% confidence interval for the sample bin mean of the inferred advection components). Vertical lines denote u*th for both $F_c$ and $F_c + S_c$, as per Figure 5.**

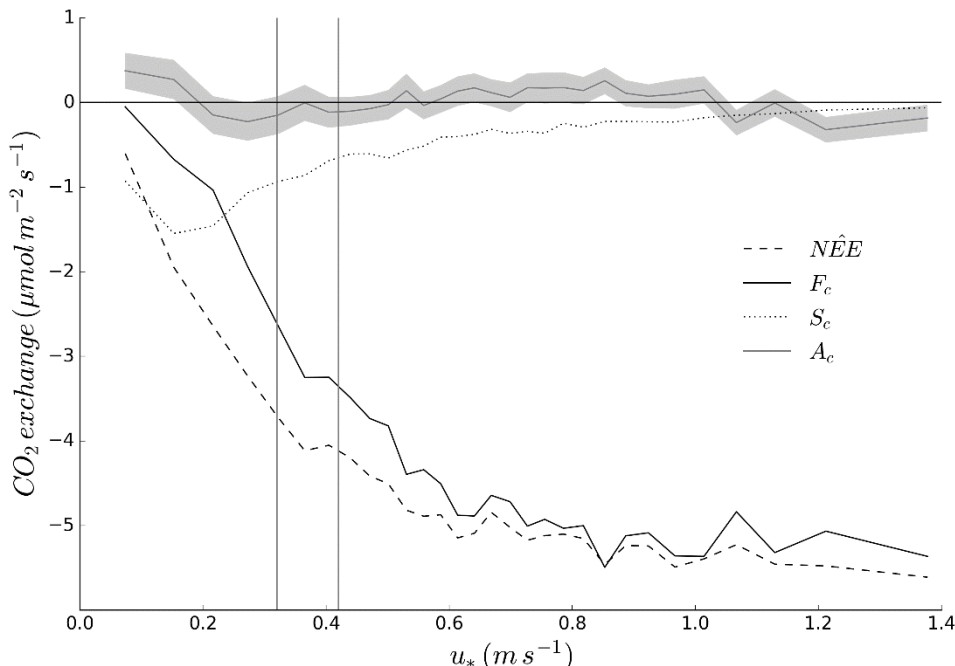

**Figure 8: dependence of measured ($F_c$, $S_c$), model-estimated ($\widehat{NEE}$) and inferred ($A_c$) mass balance components on friction velocity ($u_*$; the grey shaded area represents the 95% confidence interval for the sample bin mean of the inferred advection components). Vertical lines denote u\*th for both $F_c$ and $F_c + S_c$, as per Figure 5.**

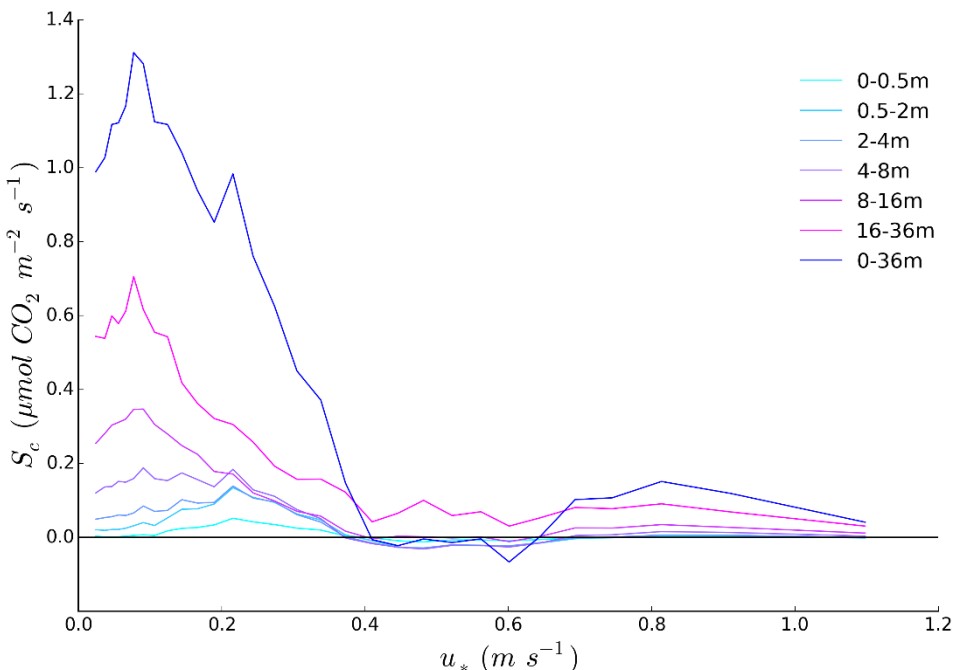

**Figure 9: dependence of measured storage components on friction velocity for individual layers.**

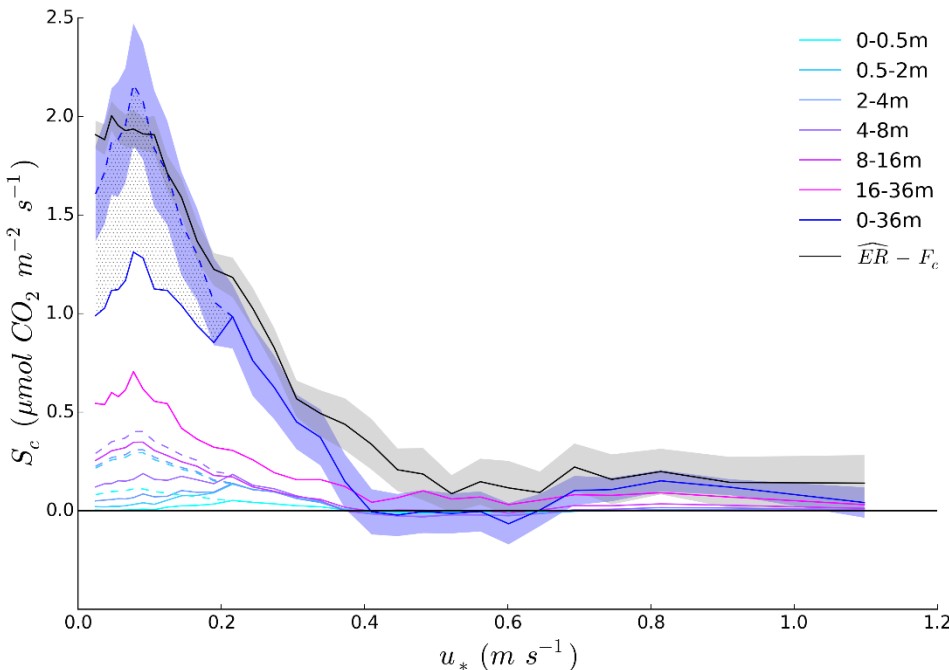

**Figure 10: dependence of corrected storage components and $\widehat{R_e} - F_c$ on friction velocity (dashed lines represent corrected storage estimates; stippled region represents difference between measured (blue) and corrected (dashed blue) 0-36m storage; shaded regions represent 95%CI for the $u_*$ bin mean).**

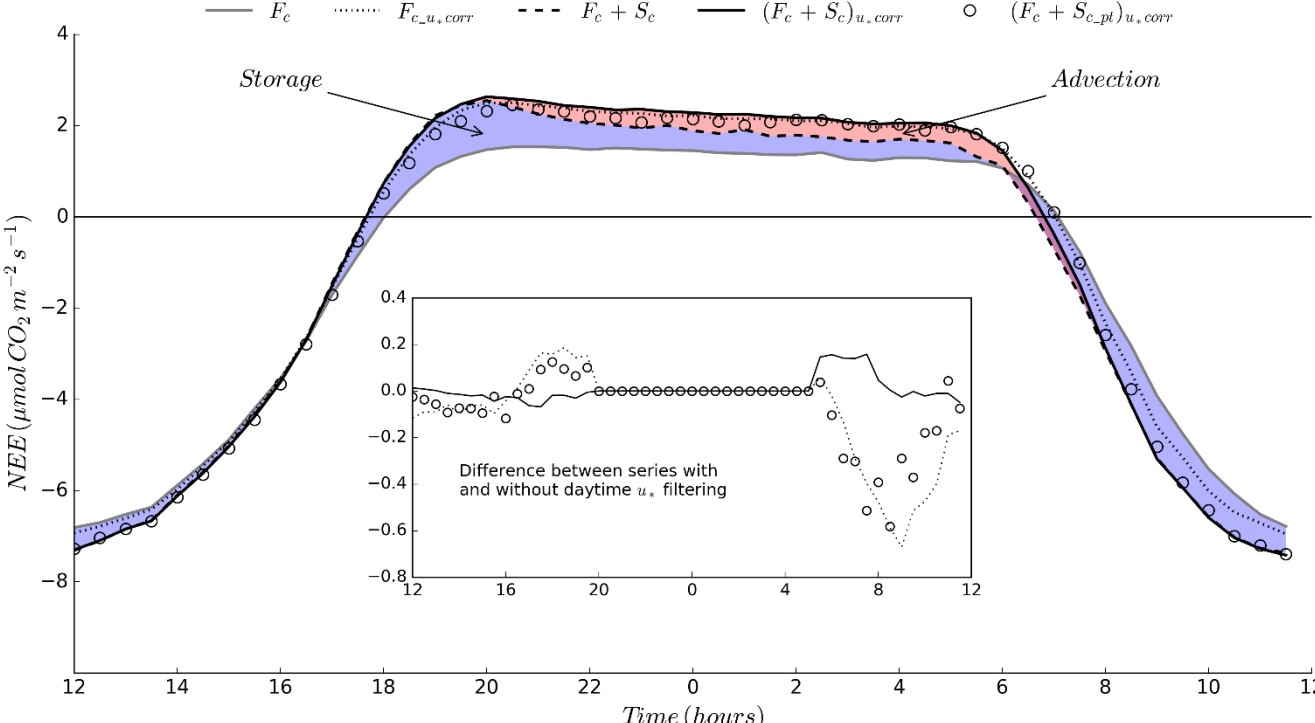

**Figure 11: effects of storage addition and u∗ filtering on diel mean NEE dynamics (inset: changes in NEE arising from application of diel versus nocturnal u∗ filter (series filtered only nocturnally subtracted from series filtered over diel cycle; note that in main plot, the u∗-corrected time series have had the diel u\* filter applied).**

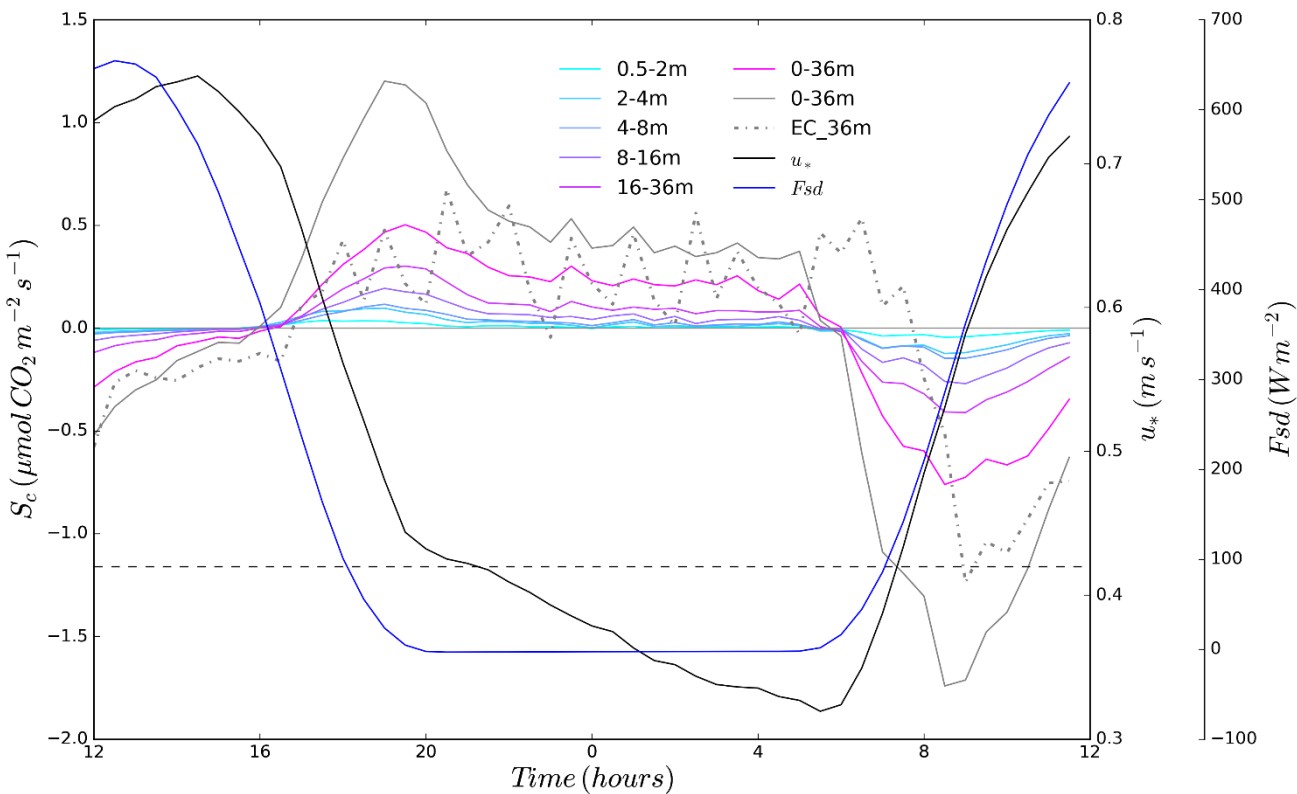

**Figure 12: mean diurnal cycle of aggregate and component $S_c$ (LH axis; horizontal solid line represents mean $S_c$) and u∗ (RH axis; horizontal dashed line represents change point-derived nocturnal u∗ threshold for $F_c$); note that day length as indicated by insolation is >12 hours due to missing data for several months during winter / spring 2013, slightly biasing the data towards longer photoperiod.**

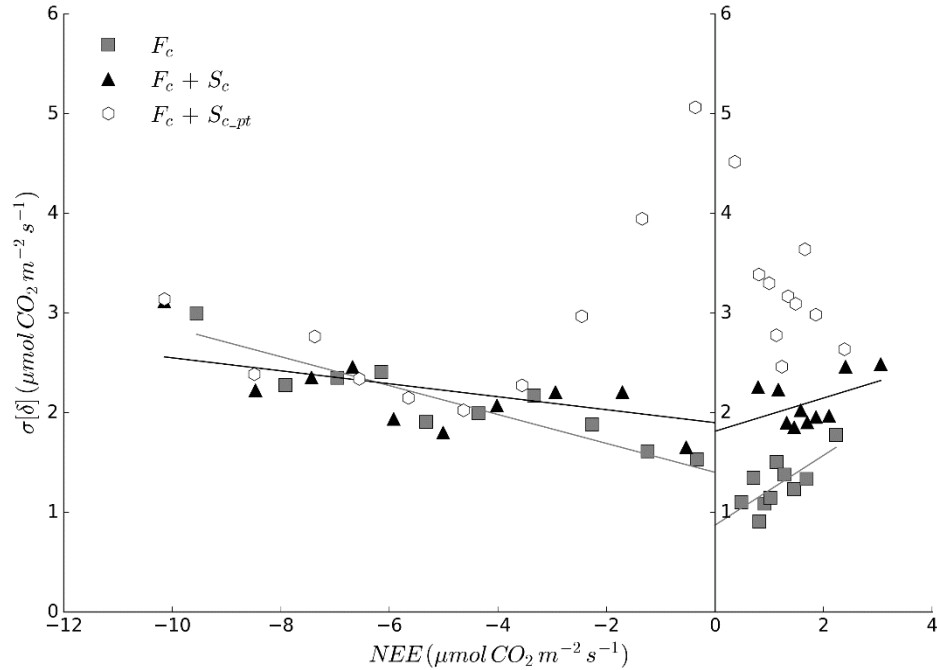

**Figure 13: standard deviation of estimated random error (σ[δ]) as a function of flux magnitude for turbulent flux (Fc), turbulent flux plus profile-based storage estimate (Fc + *Sc*) and turbulent flux plus point-based storage estimate (Fc + *Sc_pt*).**

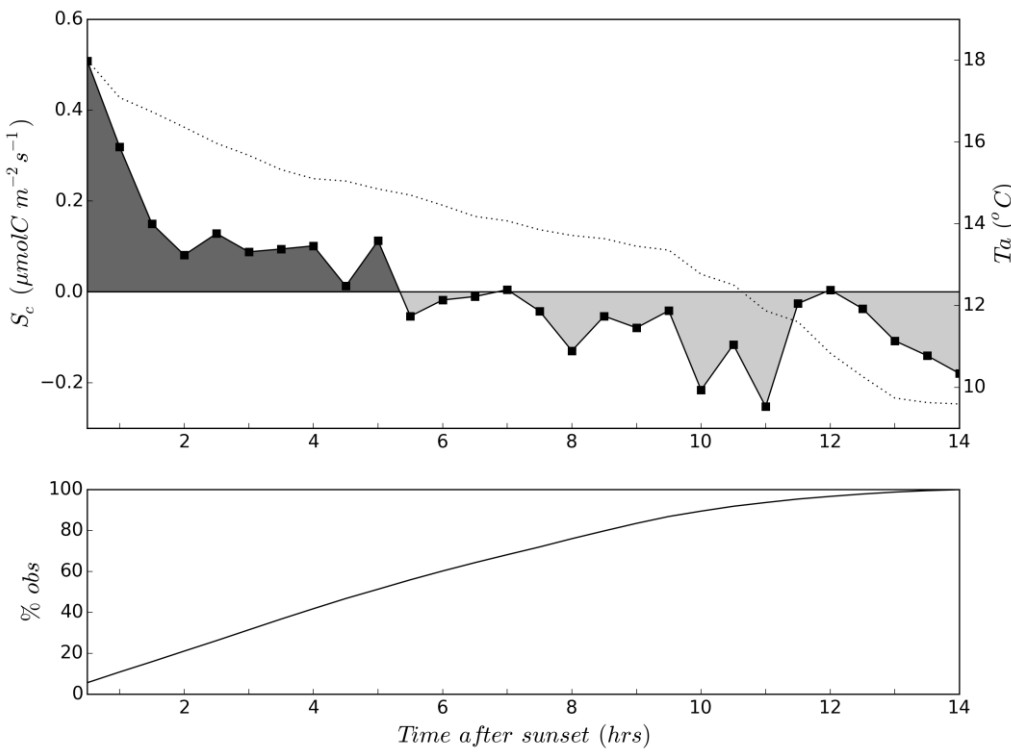

**Figure 14: dependence of $S_c$ (including only data where $u_* > u_{*th}$) on time after sunset (upper panel; dotted line is air temperature);**
5    **cumulative percentage of total nocturnal $S_c$ observations (lower panel).**

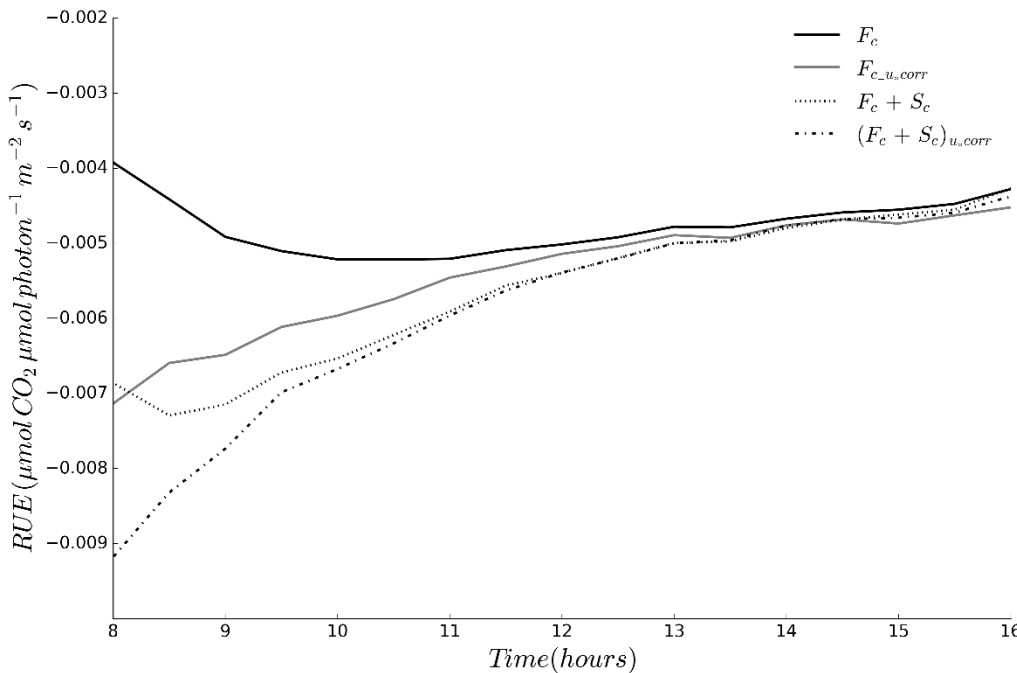

**Figure 15: effects of addition of storage term and application of daytime u∗ filtering on radiation use efficiency (RUE).**

**Table 1: site characteristics**

| | |
|---|---|
| Latitude, longitude ($^o$ dec.) | -36.673215, 145.029247 |
| Slope ($^o$) | <1 |
| Aspect | N/A |
| Dominant overstorey (>90%*) species | *Eucalyptus microcarpa* |
| Dominant understorey (>90%*) species | *Cassinia arculeata* |
| Mean canopy height ±SD (m) | 15.3±6.4 |
| Leaf area index ($m^2$ $m^{-2}$) | ~1.1 |
| Ecosystem carbon storage (tC $ha^{-1}$): | |
| • Aboveground biomass ** | 37.8 |
| • Belowground biomass *** | 10.8 |
| • Litter **** | 5.8 |
| • Soil † | 1.3 |
| Mean annual temperature ($^o$C) ‡ | 15.9 |
| Mean annual precipitation (mm): long term (1971-2000) § | 560 |

\*     By biomass (although also by number of individuals in the case of the overstorey)

\*\*    Determined from tree surveys in combination with allometric relationships developed for the relevant species in the immediate vicinity (see Paul et al., 2013 for further details).

5   \*\*\*  Determined from allometric equations relating above- to belowground biomass (see Paul et al., 2014 for further details).

\*\*\*\* Estimated from direct field survey and conversion of biomass to carbon using carbon:biomass ratio of 0.45 (Chapin et al., 2002).

†     Determined from direct survey of soils (to 40cm depth) and subsequent laboratory analysis (see Cunningham et al., 2015 for further details).

10   ‡     Determined from site observational data for years 2012-2014.

§     From nearest long-term rainfall measurement site (Mangalore Airport; Bureau of Meteorology station ID 088109).

**Table 2: site instrumentation**

| Measurement | Instrument | Manufacturer |
| --- | --- | --- |
| Wind vectors / virtual temperature | CSAT3 | Campbell Scientific Instruments |
| Radiation components | CNR4 | Kipp and Zonen |
| $CO_2$ mole fraction (eddy covariance) | LI7500 | Licor Biosciences |
| $CO_2$ mole fraction (profile) | LI840 | Licor Biosciences |
| Temperature / humidity | HMP45C | Vaisala |
| Wind speed / direction (profile) | Wind Sentry Set | RM Young |
| Barometric pressure | PTB110 | Vaisala |
| Volumetric soil water content | CS616 | Campbell Scientific Instruments |
| Soil heat flux | HFP01 | Hukseflux |
| Soil temperature | TCAV | Campbell Scientific Instruments |
| Data logging | CR3000 | Campbell Scientific Instruments |

**Table 3: statistical ranking of candidate models for ER estimation (Mode refers to the temporal fitting method, k is the number of parameters, RMSE is root mean square error, r2 is coefficient of determination, AIC is Akaike's Information Criterion, $\Delta_{AIC}$ is the AIC difference relative to the best candidate, and w is Akaike weight; note: additional candidate models were tested but were excluded if parameter estimation failed).**

| Model | | Mode | k | RMSE | $r^2$ | AIC | $\Delta_{AIC}$ | w |
|---|---|---|---|---|---|---|---|---|
| Temperature | Soil moisture | | | | | | | |
| LT | SIG | Annual+7d | 169 | 1.7391 | 0.1765 | 40732.7 | 0 | 0.83 |
| NL | SIG | Annual+7d | 169 | 1.7394 | 0.1760 | 40735.8 | 3.2 | 0.17 |
| LT | - | Annual+7d | 163 | 1.7434 | 0.1717 | 40783.9 | 51.2 | 0 |
| L | SIG | Annual | 12 | 1.7860 | 0.1309 | 40963.4 | 230.7 | 0 |
| LT | SIG | All | 4 | 1.7915 | 0.1252 | 41011.2 | 278.5 | 0 |
| LT | SIG | Annual | 12 | 1.7910 | 0.1258 | 41021.5 | 288.8 | 0 |
| NL | SIG | Annual | 12 | 1.7967 | 0.1201 | 41086.5 | 353.8 | 0 |
| NL | SIG | All | 4 | 1.7984 | 0.1184 | 41089.9 | 357.3 | 0 |
| L | SIG | Annual+7d | 169 | 1.7763 | 0.1458 | 41166.5 | 433.9 | 0 |
| L | SIG | All | 4 | 1.8147 | 0.1027 | 41274.5 | 541.8 | 0 |
| LT | - | Annual | 6 | 1.8240 | 0.0933 | 41394.5 | 661.8 | 0 |
| L | - | All | 2 | 1.8371 | 0.0801 | 41521.8 | 789.1 | 0 |
| LT | - | All | 2 | 1.8374 | 0.0798 | 41524.5 | 789.8 | 0 |
| LT2 | - | All | 4 | 1.8373 | 0.0799 | 41527.5 | 794.8 | 0 |
| LT | NL | All | 6 | 2.9857 | 0.0101 | 51472.0 | 10739.3 | 0 |
| LT2 | SIG | All | 6 | 2.9858 | 0.0038 | 51476.0 | 10743.3 | 0 |

**Candidate models**

| Origin | Code | Control | Form |
|---|---|---|---|
| Generic | L | T | $aT + b$ |
| Generic | NL | T | $ae^{bT}$ |
| Lloyd and Taylor (1994) | LT | T | $rb\, e^{E_o\left(\frac{1}{T_{ref} - T_0} - \frac{1}{T - T_0}\right)}$ |
| | LT2 | T | $rb_s\, e^{E_o\left(\frac{1}{T_{ref} - T_0} - \frac{1}{T_s - T_0}\right)} + rb_a\, e^{E_o\left(\frac{1}{T_{ref} - T_0} - \frac{1}{T_a - T_0}\right)}$ |
| Richardson et al. (2007) | SIG | VWC | $\dfrac{1}{1 + e^{(\theta_1 - \theta_2 VWC)}}$ |
| Clement et al. (2012) | NL | VWC | $e^{-0.5(\lvert\theta_v - \theta_{v0}\rvert)^2}$ |

**Table 4: lower 95%CI bound (μ - 2σ), mean (μ), and upper 95%CI bound (μ + 2σ) of Gaussian PDF of $u_{*th}$ (derived from change point detection of bootstrapped samples - see Methods), data percentile (i.e. percentage data excluded for each $u_{*th}$) and resulting imputed annual estimate of NEE. Note: i) μ – 2σ set to zero if < 0 (e.g. $F_c + S_c$ in 2013); ii) respiration and light response function analysis could not find a solution for $F_c + S_c$ in 2013 when $u_* = 0.73$ (insufficient data for robust statistical fit).**

| Year | Condition | $F_c$ | | | $F_c + S_c$ | | |
|------|-----------|-------|--|--|-------------|--|--|
| | | $u_*$ (m s$^{-1}$) | Nocturnal data percentile | Annual NEE (gC m$^{-2}$ a$^{-1}$) | $u_*$ (m s$^{-1}$) | Nocturnal data percentile | Annual NEE (gC m$^{-2}$ a$^{-1}$) |
| 2012 | μ - 2σ | 0.26 | 47 | -356 | 0.01 | <1 | -451 |
| | μ | 0.39 | 60 | -333 | 0.30 | 50 | -380 |
| | μ + 2σ | 0.52 | 72 | -337 | 0.59 | 77 | -385 |
| 2013 | μ - 2σ | 0.19 | 38 | -321 | 0 | 0 | -385 |
| | μ | 0.40 | 61 | -287 | 0.32 | 53 | -326 |
| | μ + 2σ | 0.61 | 79 | -290 | 0.73 | 87 | - |
| 2014 | μ - 2σ | 0.23 | 43 | -478 | 0.02 | <1 | -547 |
| | μ | 0.42 | 63 | -441 | 0.32 | 53 | -486 |
| | μ + 2σ | 0.61 | 79 | -445 | 0.62 | 80 | -484 |

**Table 5: gap-filled annual NEE (gC m$^{-2}$ a$^{-1}$) for 2012-2014 obtained following different data treatment ($F_c$: turbulent flux only; $F_{c\_u*\_corr}$: turbulent flux with low $u_*$ conditions removed; $F_c + S_{c\_pt}$: summed turbulent flux and point-based storage estimate; $(F_c + S_{c\_pt})_{u*\_corr}$: summed turbulent flux and point-based storage estimate with low $u_*$ conditions removed; $F_{c\_}S_c$: summed turbulent flux and profile-based storage estimate; $(F_c + S_c)_{u*\_corr}$: summed turbulent flux and profile-based storage estimate with low $u_*$ conditions removed).**

| Year | $F_c$ | | | $F_c + S_c$ | | | $F_c + S_{c\_pt}$ | | |
|------|-------|--|--|-------------|--|--|-------------------|--|--|
| | raw | $u_{*noct}$ | $u_{*24hr}$ | raw | $u_{*noct}$ | $u_{*24hr}$ | raw | $u_{*noct}$ | $u_{*24hr}$ |
| 2012 | -463 | -301 | -333 | -446 | -383 | -380 | -490 | -376 | -396 |
| 2013 | -402 | -266 | -287 | -387 | -337 | -326 | -461 | -352 | -368 |
| 2014 | -573 | -394 | -441 | -551 | -480 | -486 | -584 | -427 | -470 |

**Table 6: number of data available (after quality control and u∗ filtering), annual uncertainty due to random and model error for daytime and nighttime, and summed annual uncertainty (gC m$^{-2}$ a$^{-1}$) for diel u∗-filtered $F_c$ and $F_c + S_c$.**

| | Day | | | Night | | | Diel |
|---|---|---|---|---|---|---|---|
| Year | n, % | Error | | n, % | Error | | Error |
| | | Random | Model | | Rand | Mod | Total |
| $F_c + S_c$ | | | | | | | |
| 2012 | 6990, 39.8 | 8.0 | 11.5 | 3807, 21.7 | 6.1 | 16.2 | 22.3 |
| 2013 | 5616, 32.1 | 7.1 | 14.2 | 3087, 17.6 | 5.2 | 21.0 | 27.1 |
| 2014 | 6584, 37.6 | 8.3 | 13.5 | 3528, 20.1 | 5.6 | 17.4 | 24.2 |
| $F_c$ | | | | | | | |
| 2012 | 6990, 39.8 | 7.5 | 11.7 | 3807, 21.7 | 3.8 | 14.8 | 21.4 |
| 2013 | 5616, 32.1 | 6.6 | 14.4 | 3087, 17.6 | 3.3 | 18.2 | 24.3 |
| 2014 | 6584, 37.6 | 7.7 | 12.5 | 3528, 20.1 | 3.6 | 15.2 | 20.6 |