# Peer review of "Interactions between nocturnal turbulent flux, storage and advection at an 'ideal' eucalypt woodland site"

_Biogeosciences, 2016_

## Referee Comment (RC1) · Anonymous Referee #1 · 8 Jul 2016

General

This paper presents a thorough interpretation of discrepancies between eddy covariance and expected fluxes for an experimental site which would generally be considered as meeting the criteria for flux sites (flat and horizontally homogeneous). The authors make a strong case for the necessity of collecting flux 'storage' measurements to allow the proper interpretation of flux data. The analysis in the paper is well thought out but complicated because of the use of standard u\* correction approaches.

Specific

Throughout the paper:

You often using multiple character variables in your equations, eg NEE, Av\_c, rb, ER etc. This is considered bad practise because of the potential for mis-interpretation. For example Avc may be interpreted as A \* vc. I would recommend usage of subscripts and superscripts for differentiation of variable names eg F\_{NEE}, Av , R\_b, R\_E.

Page 3 lines 10-21:

I think the logic in these two paragraphs characterize the faults in the u\* analysis used in this paper. The total flux (NEE) is a combination of turbulent flux, storage and advection. If you only measure turbulent flux then you need to model the other two – either separately or combined – to estimate NEE. In the case of this paper you have measured storage so you only need to model advection. However, you have used the standard u\* correction approach which assumes that advection only occurs at night and only under a limited range of u\* values. This artificial restriction in the modelling of advection is what results in the double counting and some of the more elaborate explanations required later in the paper. A better approach would be to model the advection term for the entire diel period (see appendix "A" in Clement, Jarvis, Moncrieff, Agricultural and Forest Meteorology, Volume 153, 2012, Page 106). Taking such an approach would provide a simpler interpretation your already excellent data set.

Page 5 line 13:

Are you using the newer low heat emission LI7500 – or are you accounting for sensor associated heat flux enhancement in the WPL correction?

Page 7 line 22:

Was soil moisture not included in the respiration model because it was not measured for because there was no relationship? It seems that a 15 day window smoothing is a poor way of incorporating any soil moisture effects resulting from episodic precipitation

Page 7 line 23:

Using temperature measurements from 36 m as the driver for respiration would require
better explanation, particularly at night when there may be decoupling between the surface/canopy microclimate and the air well above the canopy. It may be that temperature profiles were homogeneous throughout the night (show average nocturnal T profile) or it may be that heterogenic respiration is negligible - which would likely further complicate the interpretations in this paper.

Page 8 line 22:

The error character used in the equation differs from that used in the text.

Page 9 line 1-6:

It is likely that the environmental data corresponding to the missing flux data are not representative of the environmental data corresponding to the available flux data, and it is likely that these environmental data may be under more extreme conditions. Your approach of removing random observational data – likely from periods when the model fits the data well – seems as though it will underestimate the model error.

Page 9 line 14-17:

Why do you assume there is a threshold u\*? It is possible, or likely, that advection is occurring under all u\* conditions- albeit more severely at small u\*. Determining a u\*th confidence interval is simply giving you a false sense of security. The only way to truly test for the presence of advection at night is to measure the soil and canopy respiration components and scale them up to verify that EC flux at high u\* matches scaled up chamber measurements – an approach that has its own limitations.

Page 10 line 29-31:

This statement is inconsistent with your use of the 36 m temperature as the primary determinant of nocturnal respiration.

Page 11 line 1-6:

Is it not equally plausible that cooling initiated at the surface and progressed upwards,

BGD
resulting in suppressed respiration as the depth of surface cooling increased. This could be verified by seeing the strength of the temperature profile changes. If surface cooling is strong they it is likely that your simple one-temperature respiration model will be incorrect and you will need a multi—layer model. If you indeed do have advection of low CO2 air into the bottom of the canopy then you likely have a situation of non-homogenous land cover- which may be (possibly) observed as directionally dependent effects on the nocturnal CO2 profiles.

Page 11 line 11:

It appears as if the 8 to 36 m layer has an exponential increase with decreasing u\* down to the level of u\*~ 0.1 at which it also appears to decrease. Implementing such an exponential increase is unlikely to result in as good a fit with your results. Perhaps I don't understand why you simply do not use the temperature response model of respiration using data from u\* greater than 0.5 to parameterize what was missing. Does it really matter if the advection is reducing turbulent flux or storage, it is still 'missing' flux.

```
Page 12 line 10: (".... and applying the uR*R correction,")
```

Which u\* correction - the one you developed using the complete profile or a new u\* correction based on only the point calculated storage?

```
Page 12 line 12-14: ( "We expect ..... a decline of corresponding magnitude in storage")
```

It seems as though this effect should not have an effect on storage. If within canopy, stored, CO2 is ventilated in the morning then surely the above canopy CO2 must see an increase - which would be represented as increased storage in the morning.

Page 15 line 14- 19:

This is a very useful point to make. This point alone justifies the need for storage measurements.
Page 16 line 14-15: ("...the uncertainty resulted in an increase in the potential uptake of carbon")

How can uncertainty result in an increase in uptake? Or do you mean the lower estimate of u\*th resulted in increased uptake?

Page 16 line 15-19:

From this section I assume that you are implying that true NEE will fall within the uncertainty of Fc + Sc while true NEE will not fall within the uncertainty of Fc alone (because Fc does not fall within the uncertainty of Fc + Sc). What is your justification for believing that the true NEE value will fall within the uncertainty estimate for Fc + Sc?

Page 16 line 31:

Can you explain why using u\* to remove observational data will reduce random error?

Page 16 line 31 and page 17 line 3:

On the first line you indicate that model error should be larger at night and on the second line you indicate that is larger during the day – which is it?

BGD

---

## Referee Comment (RC2) · Anonymous Referee #2 · 14 Jul 2016

In the last years there is an increasing awareness on the need of a more complete, accurate and standardized measurement setup to provide more reliable eddy covariance based flux estimates of matter and energy. This study is timely in this perspective, and provides relevant technical and scientific advancements. It promotes the use of direct, profile-based measurements of the storage, which is a relevant but generally neglected term in the net ecosystem exchange computation.

The paper is well written, but there are still some inaccuracies in the use of terms (e.g., carbon at the place of CO2) and possibly a couple of too speculative argumentations. Some of the last graphs and a few paragraphs can be removed for sake of conciseness. I strongly recommend this paper for publication having considered the following specific

indications.

Page (P) 3, Line (L) 16: 'when the nocturnal u* correction is applied'. There are some groups that apply the ustar correction at night only (a minority, to my knowledge), some others to the whole day. I recommend, for completeness of the information, to provide the carbon balance estimates with the use of the (uncommon) use of the night ustar correction, as it was already done, and with the 24 hours ustar correction.

P5 L19: '...change in carbon exchange...' CO2 is the main form with which carbon is exchanged from the ecosystem to the atmosphere, but it is not the only one; methane and VOCs are exchanged too. So please avoid this synecdoche here and elsewhere, including in some of the graphs (like Figure 13).

P6 L23: '...micrometeorological convention suggested by Chapin...', I believe that the micrometeorological convention was established well before than the paper from Chapin.

P10 L4: '...much higher random error in storage...'. To avoid this large random error, in the current ICOS protocol on storage flux measurements it is recommended to add air receivers along the lines if sequential sampling is performed, and to add some ramifications at the lower levels of air intakes to sample a wider portion of the control volume. The same argument of uncertainties originated by profile measurements is repeated in the conclusion, with possibly a technical mistake there: It is not the profile-based storage measurement that induces large uncertainty, but probably the used set-up and maybe the applied computational procedure.

P10 L27: 'Given...canopy'. A verb (are?) is missing in this sentence.

PP 11-14: the section 3.3 is very long and increasingly speculative; I lost progressively my interest and I have doubts about the argumentations. I recommend stopping at page 13, line 24, after '...in this study'.

Caption of Figure 4: '...LH axis) profile system...' I cannot understand.

Figure 11: consider removing.

Figure 12: I cannot understand what the authors mean with '...are here baselined to the height integrated profile...'. In any case, also this figure is not essential, consider removing.

Figure 13. Also this figure is not essential and unnecessarily complicated in my view, consider removing.

Table 1: 'Cassinia arculeata'->'Cassinia aculeata'.

---

## Author Comment (AC1) · 10 Sep 2016

Thanks to the reviewer for efforts and constructive commentary. Please find responses to specific critiques below.

Reviewer comment:

Throughout the paper: you often using multiple character variables in your equations, eg NEE, Av\_c, rb, ER etc. This is considered bad practise because of the potential for mis-interpretation. For example Avc may be interpreted as A \* vc. I would recommend usage of subscripts and superscripts for differentiation of variable names eg F\_{NEE}, Av , R\_b, R\_E.

**Response:**

All authors intending to submit to the Biogeosciences special issue agreed on a standard set of naming and sign conventions and some of the acronyms noted above (e.g. NEE and ER) are suggested as standard terms in the literature (e.g. Chapin et al., 2006). We don't think the use of F\_NEE is necessarily appropriate since it implies Flux\_NetEcosystemExchange. The only flux we are actually 'measuring' (i.e. through the approximating assumptions of eddy covariance) is the vertical turbulent flux at the measurement height. Therefore, when Fc and Sc are summed and corrected for low u\* conditions, we get an estimate of ecosystem / atmosphere CO2 exchange that is not strictly a flux. Advection has been dealt with in the literature typically as A or Ac. Given that we cannot separate the horizontal and vertical components of advection here, we will amend the manuscript to refer only to Ac, and to discuss the horizontal and vertical contributions as appropriate in the text.

Reviewer comment (Page 3 lines 10-21):

I think the logic in these two paragraphs characterize the faults in the u\* analysis used in this paper. The total flux (NEE) is a combination of turbulent flux, storage and advection. If you only measure turbulent flux then you need to model the other two – either separately or combined – to estimate NEE. In the case of this paper you have measured storage so you only need to model advection.

**Response:**

This may indicate that we have been somewhat unclear in defining the purpose and methodology of the paper, which we will duly amend. We have attempted to show that the advection component of the nocturnal CO2 mass balance can be coarsely approximated as the residual of the mass balance equation (equation 2 or 3 in the paper) because we can either measure or model all other components. It was not our intention to model the advection component but to show that the advection component is important even at one of the flattest forested sites in the Australian flux network;

BGD
our method allows us to do this. We also can estimate the proportional contributions of advection and storage to the shortfall in the turbulent flux relative to the source term (approximated through a simple model). This is important for reasons that we summarise below.

Č Reviewer comment (Page 3 lines 10-21):

However, you have used the standard u\* correction approach which assumes that advection only occurs at night and only under a limited range of u\* values. This artificial restriction in the modelling of advection is what results in the double counting and some of the more elaborate explanations required later in the paper.

Response:

There are two schools of thought about whether the u\* correction should be applied during the day (see Papale, 2006); I am hesitant to apply it during the day because the mechanisms that result in u\* dependence are almost exclusively nocturnal. Nonetheless, another reviewer has argued that we should filter both nocturnal and daytime data using the nocturnally derived u\* threshold, and we have undertaken to do so and include these results for comparison.

The u\* correction of course does assume that '... advection only occurs under a limited range of u\* values,' but surely this is precisely the point of the correction. The dependency between u\* and Fc (or more appropriately the sum of Fc and Sc) is interpreted as indicating that some of the respiratory CO2 source is diverted into the remaining terms in the mass balance (i.e. advection). Implicit in this is the assumption that above the threshold where there is no such dependency, the remaining terms in the mass balance are indeed negligible. We argue that this is a reasonable inference in the absence of reliable methods to directly measure advection. There are many grounds for criticism of the u\* correction (see for example Aubinet et al., 2012; Van Gorsel et al., 2007), but we make three observations: i) the site in question is relatively close to the eddy covariance ideal in terms of terrain and vegetation, and does not appear to be

BGD
subject to many of the problems that make application of the u\* correction approach particularly problematic; ii) the paper is not intended to delve into the nuances of the appropriateness of the u\* correction in general (it can probably be characterised as the flux community's least-worst option at present), but rather attempts to underscore the problems that arise when it is applied inappropriately, and; iii) other approaches, such as that of Van Gorsel et al. (2007) also must infer the presence of advection from the unphysical behaviour of flux and storage measurements under certain conditions, and thus the assumption that advection only occurs under a limited range of u\* values is not limited to the u\* correction approach. It is a necessary assumption.

We disagree that '... This artificial restriction in the modelling of advection is what results in the double counting.' In the absence of storage measurements, carbon is stored within the control volume nocturnally, thereby understating the efflux of CO2 from the ecosystem. In the morning, the stored carbon is either consumed by photosynthesis or 'flushed' upwards through the measurement height at the onset of turbulence; in either case, this results in an underestimation of morning carbon influx into the ecosystem equivalent to the magnitude of underestimation of the prior nocturnal efflux. This must be (approximately) so to satisfy conservation of mass. The application of the nocturnal u\* correction in the absence of storage measurements corrects for the underestimation of Fc due to the effects of BOTH storage and advection. Thus the efflux of the stored carbon is double-counted; once nocturnally via the u\* correction, and again in the morning when the effects of the stored carbon alters the magnitude of the turbulent flux. We are open to being convinced that our interpretation is not correct here, but the idea that double-counting occurs in the absence of profile measurements is not new, and has been made by Aubinet et al. (2002), Papale et al. (2006) and Aubinet et al. (2012), among others. As noted above, we also ascertain the proportional contributions of storage and advection to the nocturnal mass balance, because this is critical to the question of whether the u\* correction should be applied in the absence of storage measurements. If the advection term is larger than the storage term, then the u\* correction should be applied. But if the storage term is larger (as was the case for our
site), then it should not, because the double-counting problem noted above actually increases the bias in estimates of annual NEE relative to applying no correction.

Reviewer comment (Page 3 lines 10-21):

A better approach would be to model the advection term for the entire diel period (see appendix "A" in Clement, Jarvis, Moncrieff, Agricultural and Forest Meteorology, Volume 153, 2012, Page 106). Taking such an approach would provide a simpler interpretation your already excellent data set.

**Response:**

This appears to be a relatively complex analysis and would constitute a substantial further undertaking. While a comparison of such an approach with the more conventional  $u^*$ -filtering approach would be valuable, the current paper is intended to provide a cautionary note about the use of the  $u^*$  filtering technique by demonstrating the biases that arise when the technique is applied in the absence of storage measurements. It is not clear to us that this aim would be enhanced by introducing a different method of assessing the contributions of other terms in the control volume mass balance. Moreover, we note that the site in the above-cited paper is on a slope of approximately 7o, substantially in excess of that for the current site (

Č Reviewer comment (Page 7 line 22):

Was soil moisture not included in the respiration model because it was not measured or because there was no relationship? It seems that a 15 day window smoothing is a poor way of incorporating any soil moisture effects resulting from episodic precipitation

**Response:**

The use of an additional soil moisture response function increases the number of parameters to be fitted to the data when the signal:noise ratio in the data is already low, so we sought to avoid estimating too many parameters. There is little seasonal variation over the years in the free parameter in the Lloyd and Taylor temperature response function we used, but it adding a soil moisture function may improve the response to episodic soil wetting. We propose to run a more complex optimisation using a soil moisture response function and using AIC or BIC to select the most parsimonious model.

Reviewer comment (Page 7 line 23):

Using temperature measurements from 36 m as the driver for respiration would require better explanation, particularly at night when there may be decoupling between the surface/canopy microclimate and the air well above the canopy. It may be that temperature profiles were homogeneous throughout the night (show average nocturnal T profile) or it may be that heterogenic respiration is negligible - which would likely further complicate the interpretations in this paper.

Response:

We reported that the lowest RMSE for our respiration model was obtained when using the uppermost temperature sensor – this suggests that it is the most representative of all of the measurement heights, but the difference in RMSE between heights was VERY small. We believe that it is likely that the majority of the respiration is most likely from the vegetation given the very small quantity of carbon in the soil, but we have included nocturnal profiles of temperature and expanded discussion relevant to this point.

BGD
Reviewer comment (Page 8 line 22):

The error character used in the equation differs from that used in the text.

Response:

Amended.

Č Reviewer comment (Page 9 line 1-6):

It is likely that the environmental data corresponding to the missing flux data are not representative of the environmental data corresponding to the available flux data, and it is likely that these environmental data may be under more extreme conditions. Your approach of removing random observational data – likely from periods when the model fits the data well – seems as though it will underestimate the model error.

Response:

This is a good point, although it doesn't necessarily follow that because the conditions would be more extreme that the model would fit the data more poorly. More variance in both stimulus and response may improve the signal:noise. In any case, since the data are missing, this is a moot point. We now emphasise that our approach has limitations and may underestimate the error to some extent.

Reviewer comment (Page 9 line 14-17):

Why do you assume there is a threshold u\*? It is possible, or likely, that advection is occurring under all u\* conditions- albeit more severely at small u\*. Determining a u\*th confidence interval is simply giving you a false sense of security. The only way to truly test for the presence of advection at night is to measure the soil and canopy respiration components and scale them up to verify that EC flux at high u\* matches scaled up chamber measurements – an approach that has its own limitations.

Response:
We do not assume that there is a u\* threshold. We use change point detection to determine whether there is a change point in NEE when expressed as a function of u\*, and the algorithm determines that there is. We thus deduce by objective methodology that there is a u\* threshold. Of course it may be the case that advection occurs under all u\* conditions, but we are making an assumption (that is reasonably theoretically justified) that above the u\* threshold, the advection term makes a small contribution to the nocturnal mass balance. Again, such assumptions are not confined to the u\*filtering approach. Since we cannot measure advection directly, we are inferring its presence or absence from the behaviour (in response to meteorological conditions) of the mass balance terms we CAN measure. It seems therefore that this is a criticism that applies to the field rather than our paper alone. We also note that the profile system gives us a secondary and completely independent estimate of the u\* threshold. It seems reasonable to assume that if the storage term goes to zero under well-mixed conditions, advection (at least associated with drainage flows) is likely to be minimal. We thus argue that allowing for the possibility that the u\* threshold may be anywhere within a large uncertainty range (and propagating this to annual NEE estimates) is perhaps unduly conservative, since the profile system yields the same best estimate for u\* threshold. Nonetheless, we are happy to further emphasise that the range of uncertainties we have been able to quantify are only a subset of the true uncertainty in the NEE estimates. We also agree that chamber-based scaling up of nocturnal (and ultimately daytime) respiration would be a highly desirable addition to this (and pretty much every other eddy covariance) study, but it was beyond the resources we had available for this project. It is something we hope to do in future, and we will emphasise this in the conclusions of the paper.

Reviewer comment (Page 10 line 29-31):

This statement is inconsistent with your use of the 36 m temperature as the primary determinant of nocturnal respiration.

Response:
Agreed, and will be amended.

Reviewer comment (Page 11 line 1-6:

Is it not equally plausible that cooling initiated at the surface and progressed upwards, resulting in suppressed respiration as the depth of surface cooling increased. This could be verified by seeing the strength of the temperature profile changes. If surface cooling is strong they it is likely that your simple one-temperature respiration model will be incorrect and you will need a multilayer model. If you indeed do have advection of low CO2 air into the bottom of the canopy then you likely have a situation of non-homogenous land cover- which may be (possibly) observed as directionally dependent effects on the nocturnal CO2 profiles.

Response:

We prepared figures that show increased cooling as a function of u\* in the lower layers relative to the upper layers, but in the end excluded them because we determined that this should not logically result in declines in Sc. This is because lower temperatures imply two antagonistic effects: i) lower temperatures will result in reduced respiration, but; ii) lower temperatures at lower levels are also generally indicative of less turbulent conditions, since radiant heat loss from the surface cools the lower layers, which will will only remain so if there is limited turbulence to replenish heat from above. The limited turbulence will therefore limit the magnitude of Fc whilst increasing Sc, unless there is advection. Nonetheless, we will clarify this point in the text and include a plot showing the temperature profiles as a function of u\*. With respect to the point about non-homogeneous land cover, we expect that the flux footprint becomes sufficiently large at low u\* (stable conditions) that the flux footprint most likely extends beyond the boundaries of the ecosystem. The flow is predominantly westerly, and we consider it likely that under calm conditions, there are gravity flows from the northwest. Under these conditions, the flux footprint may extend several kilometres to the ridge, in which case there is likely entrainment of CO2-depleted air from the overlying atmosphere that BGD
results in horizontal variations in the CO2 field and corresponding advective losses. However, this is largely speculative; in the literature, most terrain-induced flows are <0.4m s-1. To measure such weak winds requires 2D sonic anemometers (generally precision of up to 0.01 m s-1). Our cup anemometers have a start-up speed of 0.5 m s-1 so we cannot measure these effects. Wind directions for the mechanical vanes are not valid once the velocity is <0.8 m s-1 (and most certainly not below 0.5m s-1, when the measured wind speed will be zero), so that a directional analysis is unlikely to be useful. Again, we can emphasise this point in the text.

Reviewer comment (Page 11 line 11):

It appears as if the 8 to 36 m layer has an exponential increase with decreasing u\* down to the level of u\* 0.1 at which it also appears to decrease. Implementing such an exponential increase is unlikely to result in as good a fit with your results. Perhaps I don't understand why you simply do not use the temperature response model of respiration using data from u\* greater than 0.5 to parameterize what was missing. Does it really matter if the advection is reducing turbulent flux or storage, it is still 'missing' flux.

**Response:**

We can amend the algorithm to include a non-linear relationship in the 16-36m layer, but we are not convinced this will make a large difference, since the correction is based on using the average of the 8-16m and 16-36m layers. Our rationale for including this element of the analysis was that the response of the storage term to u\* implies the presence of a specific advective mechanism (drainage flows, which are expected to mostly affect the lower layers, because they are typically shallow features. If our simple linear correction of the lower layer storage terms removes the decline in NEE as a function of u\* (or as reported in the paper, causes the corrected Sc to be approximately equal to the modelled source term minus the observed turbulent flux), this suggests that loss of carbon from the lower layers of the control volume is primarily responsible for the de-
cline in Fc + Sc as a function of u\*. In turn, this supports the hypothesis that drainage flows may be primarily responsible for advective losses. We acknowledge that this element is somewhat speculative (and in the paper we emphasise that instrumentation to directly measure the presence of these flows – e.g. a profile of windsonics – would be a valuable addition). Our primary motivation for its inclusion was to show that for sites with more difficult measurement conditions (e.g. hillier terrain) where a larger proportion of turbulent flux data may be compromised, it may be possible to develop a correction – or at least gain insight into the prevailing nocturnal dynamics – based on the behaviour of the storage term. We didn't emphasise this as our motivation, and will amend the text to do so (as well as noting that the additional instrumentation noted above would make such an endeavour less speculative).

Reviewer comment (Page 12 line 10):

("  $\dots$  and applying the uR\*R correction,") Which u\* correction - the one you developed using the complete profile or a new u\* correction based on only the point calculated storage?

Response:

This was not made at all clear. The analysis used u\* estimates derived from the sum of the point storage estimate and the turbulent flux. This will be made clear in the amended text.

Reviewer comment (Page 12 line 12-14):

( "We expect ... a decline of corresponding magnitude in storage") It seems as though this effect should not have an effect on storage. If within canopy, stored, CO2 is ventilated in the morning then surely the above canopy CO2 must see an increase - which would be represented as increased storage in the morning.

Response:

We are not sure how to interpret this criticism. We still expect a bias towards efflux

BGD
because the point-based estimate of Sc is smaller in the morning than the 'true' (i.e. profile-based) estimate of Sc. That being the case, it doesn't properly compensate for the effect of the nocturnally respired CO2 that is either released or consumed by photosynthesis in the morning. So we may not have double counting, but perhaps 1.5 counting! In any case, the point is that we don't consistently see this because the point-based estimates are so erroneous that they seriously affect the parameter optimisation and subsequent model estimates in unpredictable ways.

Reviewer comment (Page 15 line 14-19):

This is a very useful point to make. This point alone justifies the need for storage measurements.

Response:

We also thought this was important, and to some extent novel. However, another reviewer suggested removing the figure upon which this part of the analysis ultimately rests (Figure 13)!

Reviewer comment (Page 16 line 14- 15):

(" ... the uncertainty resulted in an increase in the potential uptake of carbon") How can uncertainty result in an increase in uptake? Or do you mean the lower estimate of  $u^*$ th resulted in increased uptake?

Response:

The latter interpretation is correct. The phrasing in this section was poor, and will be amended to improve clarity.

Č Reviewer comment (Page 16 line 15-19):

From this section I assume that you are implying that true NEE will fall within the uncertainty of Fc + Sc while true NEE will not fall within the uncertainty of Fc alone (because Fc does not fall within the uncertainty of Fc + Sc). What is your justification for believing

BGD
that the true NEE value will fall within the uncertainty estimate for Fc + Sc?

Response:

Some of the phrasing in this section is poor as noted above. We intended to argue that calculating and quoting uncertainties in the absence of storage measurements is a meaningless exercise, because the best estimate for NEE calculated from turbulent AND storage fluxes is not within this range, and there is not even any overlap in the uncertainties. We also tried to convey that our uncertainties were not definitive (i.e. we cannot know that the true NEE value lies within our uncertainty range), with the following: 'It should be emphasised that there are numerous sources of uncertainty that have not been quantified here. Perhaps most important of these is systematic errors in the measurements themselves, which may be an extremely important source of true uncertainty (Lasslop et al., 2008). Thus the uncertainties reported here for FRcR + SRcR also should not be formally interpreted as total uncertainty in the true source / sink term, but as the uncertainty contributed by a subset of quantifiable errors.' If this is inadequate we can expand the qualifications if the reviewer has further suggestions.

Reviewer comment (Page 16 line 31):

Can you explain why using u\* to remove observational data will reduce random error?

Response:

Since every observational datum contains random error, the more observational data is removed by filtering, the more random error is reduced.

Reviewer comment (Page 16 line 31 and page 17 line 3):

On the first line you indicate that model error should be larger at night and on the second line you indicate that is larger during the day – which is it?

Response:

Poor wording again. The first statement is meant to explain that the nocturnal model
error is large relative to the nocturnal random error because much of the observational data is filtered out by the u\* filter at night. The second statement is meant to say that the daytime model error is comparable to the nocturnal model error because although there is less missing data during the day, random nocturnal error affects the parameter estimation for ER, which then propagates to the day because these parameter estimates are used in the NEE calculation. This will be clarified in plain English. Also, we have realised that a table that should contain the random and model uncertainty estimates was left out of the manuscript. This will be reinserted.

---

## Author Comment (AC2) · 10 Sep 2016

Thanks to the reviewer for efforts and constructive critique. Please find responses to specific items below.

Reviewer Comment: The paper is well written, but there are still some inaccuracies in the use of terms (e.g., carbon at the place of CO2) and possibly a couple of too speculative argumentations. Some of the last graphs and a few paragraphs can be removed for sake of conciseness. I strongly recommend this paper for publication having considered the following specific indications.

Response:

See specific points below.

Reviewer comment (P3, L16):

'when the nocturnal u\* correction is applied'. There are some groups that apply the ustar correction at night only (a minority, to my knowledge), some others to the whole day. I recommend, for completeness of the information, to provide the carbon balance estimates with the use of the (uncommon) use of the night ustar correction, as it was already done, and with the 24 hours ustar correction.

Response:

We have rerun analyses using both cases.

Reviewer comment (P5 L19):

"... change in carbon exchange..." CO2 is the main form with which carbon is exchanged from the ecosystem to the atmosphere, but it is not the only one; methane and VOCs are exchanged too. So please avoid this synecdoche here and elsewhere, including in some of the graphs (like Figure 13).

Response:

We amend all references to carbon to CO2 where appropriate throughout.

Reviewer comment (P6 L23):

"... micrometeorological convention suggested by Chapin ...' I believe that the micrometeorological convention was established well before than the paper from Chapin.

Response:

Reference to Chapin has been removed.

Č Reviewer comment (P10 L4):

'Much higher random error in storage...' To avoid this large random error, in the current
ICOS protocol on storage flux measurements it is recommended to add air receivers along the lines if sequential sampling is performed, and to add some ramifications at the lower levels of air intakes to sample a wider portion of the control volume. The same argument of uncertainties originated by profile measurements is repeated in the conclusion, with possibly a technical mistake there: It is not the profile-based storage measurement that induces large uncertainty, but probably the used setup and maybe the applied computational procedure.

**Response:**

It is likely that our setup increases the random error in the profile data relative to that described by the reviewer. Our system can sample the entire 36 m profile in 2 minutes because air is drawn through all lines simultaneously, although this reduces pump speed and results in a lag from intake to analysis of slightly greater than 1 m. We calculate the storage term as the difference between the height-integrated CO2 molar density between the beginning and end of the half-hourly period corresponding to the flux averaging period. For example, we use the 14:30-14:32 interval and the 15:00-15:02 interval to calculate the storage term for the 15:00 flux estimate, which represents 14:30-15:00. The 2-minute lag is introduced to account for the system lag between intake and IRGA.

Adding upstream volume to the system (i.e. a buffer chamber at the intake) would be expected to reduce random error by smoothing near-instantaneous fluctuations in [CO2], but this is effectively equivalent to using a longer time average. We can use a longer time average because we measured all 2-minute periods and can use a 2, 4 or 8 minute period at the beginning and end of the half-hour. However, according to Finni-gan (2006): `... there is an irreducible error associated in calculating the storage from a single tower, where the worker must choose between the random error associated with using instantaneous profiles and a certain loss of high frequency information if the storage term is calculated from time-averaged vertical profiles.'
For this reason, we think that our choice of a single 2-minute period is defensible. We believe that the best-designed profile system will increase the amount of random error observed when Fc is summed with Sc compared with Fc alone. Thus, we do not entirely agree that 'It is not the profile-based storage measurement that induces large uncertainty, but probably the used setup and maybe the applied computational procedure.' Our setup may increase random error over and above the 'ideal' profile system, but random error will not be eliminated from the ideal system. That said, a different intake design in which there is some horizontal spatial sampling would further reduce random fluctuations, and our system does not have this (single inlet). Therefore, we acknowledge this as a limitation of the design, and accordingly make conclusions on this aspect of the analysis less general.

Reviewer comment (P10 L27):

'Given ... canopy'. A verb (are?) is missing in this sentence.

Response:

Amended.

```
Reviewer comment (PP 11-14):
```

The section 3.3 is very long and increasingly speculative; I lost progressively my interest and I have doubts about the argumentations. I recommend stopping at page 13, line 24, after '... in this study'.

Response:

Acknowledged. Section 3.3 will be shortened and speculative aspects removed.

Reviewer comment:

Caption of Figure 4: '... LH axis) profile system ...' I cannot understand.

Response:
Amended so that description of data precedes the indication of which axis it is associated with (Figure 4: CO2 mole fraction time series (11/02/2012 - 19/02/2012) for all profile heights (LH axis) and corresponding friction velocity time series (RH axis).

Reviewer comment:

Figure 11: consider removing.

Response:

This figure demonstrates that the addition of the profile data to the flux data substantially increases the nocturnal random error relative to the flux data alone, and also the very large error associated with the point storage method when fluxes are small. This seems important given that we dismiss the point-based storage estimates partially on the basis that the model parameterisation becomes unstable when using these data (this figure illustrates why!). If the reviewers consider that such information can be stated in the text without recourse to a figure, we will remove it.

Reviewer comment:

Figure 12: I cannot understand what the authors mean with '... are here baselined to the height integrated profile... '. In any case, also this figure is not essential, consider removing.

Response:

This figure pertains to some of the more speculative aspects in section 3.3 critiqued above. Will be removed as part of the process of clarifying and shortening section 3.3.

Č Reviewer comment:

Figure 13. Also this figure is not essential and unnecessarily complicated in my view, consider removing.

Response:
We considered this a novel finding. It is often argued that in the absence of profile measurements, nocturnal measurements with  $u^* > u^*$  threshold immediately following low  $u^*$  conditions should be filtered out because they result in an overestimation of ER due to the release of stored carbon. We show here that while this occurs late at night (as indicated by the negative storage term >5 hours after sunset in Figure 13), in the early evening, the storage term is positive, and NEE is therefore underestimated. This has implications for the calculation of relationships between temperature and ER, an in turn for the extrapolation of these data to the daytime (as is typically done as part of the partitioning process). It may perhaps be that this finding is not made sufficiently clear in the discussion, or it may be that it is not considered relevant by the reviewer. If the latter, we can potentially remove this analysis and publish in a separate paper.

Reviewer comment:

Table 1: 'Cassinia arculeata'->'Cassinia aculeata'.

Response:

Amended.

---

## Author Response (AR1)

Response to referee comments

Referee 1

| Suggestion | Action |
|---|---|
| I do, however, find the text to be a bit too 'wordy' with the descriptions of the processes involved taking up too much text and being hard to follow. | Introduction and conclusions simplified and shortened. The main body otherwise stands largely unchanged; while we are happy to accept and respond to further criticism of this section, I hope we can respond appropriately to more specific critique in the next phase of review. |
| I suggest that the authors present their conceptual model of the relevant components (eg source = [turbulent flux] + [turbulent flux lost as advection] + [storage] + [storage lost as advection]) so that it is easier to present these concepts in the text. | We are not able to quantify the suggested terms using the method outlined in the paper. All we are able to ascertain is the partitioning of the source term between advection, turbulent flux and storage. This probably underscores that the original manuscript did not provide sufficient clarity in explaining the approach. Again, hopefully this is something that can be amended in full review. |

Referee                                                   2                                          2

| Suggestion | Action |
|---|---|
| I recommend to change in the y axis of several figures (e.g. figure 14, micromol C into micromol CO2). Reason: the mole unit should be referred to molecules, and C alone is not a molecule. | All instances amended in Figures |
| I also recommend to change the somewhat confusing term 'mixing ratio' into 'dry molar fraction', if that was the unit the Authors wanted to use. | All instances amended in text |

[revised manuscript text omitted]
 (i.e. a negative change in storage balancing the vertical turbulent transfer of the nocturnally accumulated carbon), the requirement for the storage term to be approximately zero is violated, and the nocturnally respired carbon is effectively counted twice (Aubinet et al., 2002). This unavoidably biases measurements towards net carbon efflux, and also affects the apparent relationship between ecosystem carbon fluxes and climatic controls.

Given the number of sites that do not have profile systems, it is thus important to quantify the effects of failing to measure storage. In this study, we use a three year-record of carbon exchange (including storage measurements) for an Australian eucalypt woodland to investigate the interaction between nocturnal turbulent flux, storage and advection. We devise a simple method to infer the magnitude of advection and in turn quantify the apportionment of the nocturnal ecosystem carbon source between turbulent flux, storage and advection. We quantify the biases in annual carbon exchange that arise from neglecting the storage term and discuss its effects on interpretation of carbon fluxes in the context of climatic drivers. Given the significant additional investment and complexity associated with the construction and deployment of profile systems alongside eddy covariance systems, it might be argued that the incurred bias of neglecting storage could be ignored if it is small relative to other measurement uncertainties. We therefore also propagate the errors associated with determination of $u_{*th}$, random measurement error and imputation (gap-filling) error to annual estimates of net carbon exchange, and assess their magnitude relative to biases due to neglect of storage.

Over the past 2 decades, eddy covariance measurements have been widely adopted as a tool for aggregate flux measurement (Baldocchi, 2003), and there are now over 650 operational monitoring sites registered with the international flux network (Fluxnet: fluxnet.ornl.gov). Within the Australian regional network (OzFlux: www.ozflux.org.au), there are 29 active sites (Beringer et al., 2016, this issue). The use of the eddy covariance technique allows continuous automated monitoring of mass and energy fluxes, and long-term multi-site datasets have yielded valuable ecological insights in recent years (Baldocchi, 2008).

The eddy covariance technique requires a number of implicit simplifying assumptions, perhaps the most crucial of which is that the vertical flux across the horizontal measurement plane at a given height above the surface approximates the true environmental source / sink of the measured entity. Only under a limited range of atmospheric (vigorous turbulence and approximate statistical stationarity of vector and scalar quantities) and surface conditions (level terrain and upwind horizontal homogeneity) is this assumption strictly met (Kaimal and Finnigan, 1994). When these conditions are not met, the turbulent flux at the measurement height may be unrepresentative of surface exchange, as storage of the entity below the measurement height or mean transport (advection) associated with the development of horizontal or vertical gradients in the scalar fields (assuming negligible flux divergence) occurs (Aubinet et al., 2012). This situation primarily arises nocturnally when surface radiative cooling causes the collapse of buoyancy-generated turbulence. It has long been recognised (e.g. Goulden et al., 1996b) that this is particularly problematic for the measurement of ecosystem carbon exchanges because there is substantial nocturnal respiratory $CO_2$ efflux at from soil and vegetation. If this efflux is underestimated, then the net carbon exchange over aggregating periods > 24 hours is correspondingly biased towards surface uptake.

While the sum of turbulent flux and storage may be sufficient to characterise the surface flux at some sites (e.g. Dolman et al., 2002), nocturnal advection is thought to occur to varying extents at most flux sites (Aubinet, 2008), primarily via horizontal gravity-induced terrain drainage flows that occur on slopes of as little as 1° (Aubinet et al., 2003; Staebler and Fitzjarrald, 2004). Moreover, length rather than gradient of slope may be more critical for the initiation of such flows (Finnigan, 2008). Most non-agricultural ecosystem measurement sites are on sloped terrain because historically, flat, arable land has been cleared for agriculture. Drainage-induced horizontal advection is thought to be among the most important source of systematic nocturnal measurement error (Aubinet et al., 2012).

Up to the present, attempts to measure advection directly have been complex and resource-intensive and have yielded highly uncertain results (Aubinet et al., 2010; Feigenwinter et al., 2008; Leuning et al., 2008). Thus indirect approaches to nocturnal data correction are generally applied. Most common is the identification of a threshold of turbulence (using friction velocity – herein $u_*$ – as a measure of mean turbulent activity) below which the nocturnal turbulent carbon flux declines with $u_*$ (Goulden et al., 1996b). Given that there should be no functional relationship between ecosystem respiration and $u_*$, this decline – in the absence of corresponding declines in the factors that control respiration – is interpreted as an increase in the

storage and advection terms at the expense of the turbulent flux. Data below this threshold (herein $u_{*th}$) are discarded and replaced using functional relationships between known physical respiratory drivers (primarily temperature) and nocturnal carbon fluxes, optimised for periods when $u_* > u_{*th}$ (herein this approach is referred to as the $u_*$ correction).

5    The $u_*$ correction has been criticised on theoretical and practical grounds (Aubinet et al., 2012; Aubinet and Feigenwinter, 2010; Van Gorsel et al., 2007), but remains the most widely adopted approach to nocturnal data correction. The $u_*$ correction should be applied to the sum of the measured turbulent flux and storage terms (Papale, 2006), yet only 10-30% of sites have deployed profile systems to measure storage (Papale, pers. comm., 23/11/2015). In Australia, only four of the 29 active sites have profile measurements, whereas $\geq$ 15 sites have canopies of sufficient height to warrant them (using
10   measurement height > 3 m as a threshold for requirement). The storage term (which is proportional to the time rate of change of $CO_2$-carbon density) is approximately zero over the long term (e.g. annually) because turbulent mixing ensures the ambient $CO_2$ mixing ratio is (neglecting anthropogenic addition) approximately stable over time. As such, it may appear that neglect of the storage term would not affect annual carbon balances. However, this is not the case, as discussed below.

15   The storage term is positive at night because the $CO_2$ mixing ratio in the layer between the eddy covariance measurement level and the ground rises as respired carbon accumulates due to the decline of turbulent transfer. Following sunrise, the sign of the storage term reverses. This occurs through two processes. First, insolation-driven surface heating re-establishes buoyancy-generated turbulent activity, and so, assuming horizontal homogeneity, drives vertical turbulent transfer of nocturnally respired carbon upwards through the measurement plane, thereby reducing the $CO_2$ mixing ratio below the
20   plane. The second process is net photosynthetic drawdown of carbon. For several hours after sunrise, the storage term reflects the superimposed effects of both processes, but it is the first process that is problematic when the u* correction is applied without storage measurements.

     Since the $u_*$ correction removes all nocturnal data where the turbulent flux declines as a function of $u_*$, it is implicitly
25   correcting for the combined effects of nocturnal advection *and* storage. However, the turbulent transfer of nocturnally respired carbon after sunrise is embedded in the turbulent flux signal, so is counted twice (Aubinet et al., 2002). However, mass conservation dictates that the morning turbulent release of the nocturnally respired carbon must be balanced by an equivalent change in storage. Therefore, the sum of the turbulent flux *and* storage terms is required to approximate the real time carbon source / sink. This does not imply that the $u_*$ correction should be avoided at sites where storage measurements
30   are not available because it is necessary to correct for nocturnal advective carbon losses. The amount of bias following application of the $u_*$ correction at such sites depends on the relative effects of storage and advection on the nocturnal carbon balance. This implies that if the effects of advection dominate in reducing nocturnal turbulent carbon flux, then the correction is advisable because most of the carbon that would otherwise be ventilated in the morning has been removed; however, if the dominant process is storage, the correction should be avoided.

~~There is a need to understand the implications of neglecting the storage term given the number of sites that do not have profile systems. These implications include the likely long-term biases in ecosystem / atmosphere carbon exchanges but also the distortion of relationships between carbon exchange and key environmental controls of exchange processes. Moreover, it is important to quantify the contributions of advection and storage to nocturnal flux underestimation to understand the implications of applying the $u_*$-correction at a given site. In this study, we used a three year record of carbon exchange (including storage measurements) for an Australian eucalypt woodland to investigate the interaction between nocturnal turbulent flux, storage and advection. We devised a simple method to infer the influence of advection and we quantify the biases in annual carbon exchange that arise from neglecting the storage term and the complications that arise with respect to ecophysiological interpretation of flux data. Given the substantial cost of profile systems (the system described subsequently cost approximately $17000AUD), 
[revised manuscript text omitted]

Here, NEE (net ecosystem exchange of carbon) is the true source term, term $I$ is the turbulent flux across the upper horizontal plane of the control volume at instrument height $h_m$ (**w** is the vertical velocity, and overbar and prime denote mean and quasi-instantaneous fluctuation from mean, respectively), term $II$ is the storage term integrated over finite time period ($t$) and control volume depth ($z$), term $III$ is the sum of the advection components in the horizontal dimensions ($x$ and $y$, with corresponding vectors **u** and **v**) and term $IV$ the vertical advection. We adopted the standard micrometeorological convention suggested by Chapin et al. (2006) in which NEE is positive (negative) when the net transfer of carbon is from ecosystem to atmosphere (atmosphere to ecosystem). In the following text, equation 2 is simplified to:

$$NEE = F_c + S_c + Av_c + Ah_c \tag{3}$$

Here, $F_c$ and $S_c$ are the turbulent flux and storage terms, and $Av_c$ and $Av_h$ are the vertical and horizontal advection terms, respectively. During the day, when turbulence is well-developed, the turbulent flux ($F_c$) is generally the dominant term, but at night, the other terms may become dominant under weak mixing. Following the identification of $u_{*th}$, nocturnal data were rejected where $u_* < u_{*th}$. While nocturnal advection was not measured, it was inferred as the residual of the terms in equation 3. Nocturnally, NEE is equivalent to ecosystem respiration (herein ER). While ER is unknown when $u_* < u_{*th}$, it can be estimated ($\widehat{ER}$) using an empirical model, the parameters of which are optimised for periods in which the sum of turbulent flux and storage approximates ER (*i.e.* when $u_* > u_{*th}$). Equation 3 thus becomes:

$$\widehat{ER} - F_c - S_c = Av_c + Ah_c \tag{4}$$

**2.5 Imputation**

Carbon flux and temperature data were used to optimise the parameters of an empirical temperature response function (optimisation used the Levenberg Marquardt algorithm implemented in the Python Scipy package) that was then used to estimate ER for $u_* < u_{*th}$ (and for subsequent gap-filling). The model was an Arrhenius-style function proposed by Lloyd and Taylor (1994):

$$ER = rb\, e^{E_o\left(\frac{1}{T_{ref} - T_0} - \frac{1}{T - T_0}\right)}. \tag{5}$$

Here, $rb$ is the reference respiration at a reference temperature ($T_{ref}$), $E_o$ is an activation energy parameter that controls temperature sensitivity, and $T_0$ is the temperature at which metabolic activity approaches zero. $T_0$ and $T_{ref}$ are fixed at

–46.02 °C and 10 °C, respectively (Lloyd and Taylor 1994); the unconstrained version of the function is overparameterised (Reichstein et al., 2005; Richardson and Hollinger, 2005). We fitted $E_o$ annually and then derived *rb* for successive 15-day windows (5-day step with linear interpolation to generate a continuous time series) (Reichstein et al., 2005). This avoids confounding of diurnal and seasonal temperature responses, and allows the model to capture low-frequency variation in ER associated with variables not explicitly represented in the model (*e.g.* soil moisture and substrate availability). We used temperature measured at the EC height (36 m) as the respiratory driver because it was found to have the lowest RMSE (relative to soil – which had the highest - and 0.5, 2, 4, 8 and 16 m air temperatures; data not shown).

Gap-filling was also required for daytime to assess the effects of nocturnal data treatment on annual NEE. We used a Michaelis Menten-type rectangular hyperbolic model (Ruimy et al., 1995) of modified form (Falge et al., 2001) to estimate NEE, where ER was calculated from equation 5 using daytime temperatures in conjunction with nocturnally-derived parameter estimates,:

$$NEE = \frac{\alpha Q}{1 - Q/2000 + \alpha Q/\beta} + ER. \tag{6}$$

Here, $\alpha$ is the initial slope of the photosynthetic light response, $Q$ is photosynthetic photon flux density, and $\beta$ is photosynthetic capacity at 2000 μmol photons $m^{-2} s^{-1}$. The same window size, window step and interpolation procedure was used as for the nocturnal fitting of the respiration model. We adopted the additional light-response model criterion in which $A_{opt}$ is modified to include a non-linear scaling factor to account for the effects of vapour pressure deficit (VPD) on stomatal conductance (Lasslop et al., 2010):

$$\beta = \begin{cases} \beta_0 \, e^{(-k(VPD - VPD_0))}, & VPD > VPD_0 \\ \beta_0, & VPD < VPD_0 \end{cases} \tag{7}$$

Here, $VPD_0$ is a threshold value above which stomatal conductance becomes sensitive to VPD, and $k$ is a fitted parameter defining the $\beta$ response to VPD.

**2.6 Uncertainty estimation**

[revised manuscript text omitted]

20   this represents carbon losses associated with the remaining mass balance terms (*i.e.* advection). This can be quantitatively estimated as a residual following equation 4. Note that parameter optimisation of the temperature response function used $F_c + S_c$ as the target variable because $S_c$ is observed to be non-zero when $u_* \gg u_{*th}$ (see Figure 3, and subsequent discussion).

25   The terms in equation 4 are plotted as a function of $u_*$ in Figure 5. The inferred advection estimate increased rapidly below $u_*$ threshold $= 0.32$ m s$^{-1}$, and was comparable to $S_c$ at the lowest $u_*$ values. This indicates that under the calmest conditions, $F_c$ approached zero, and approximately half of the carbon respired by the ecosystem was stored below the measurement height while the remainder was advected away. Integrated over the interval $0 < u_* < u_{*th}$, $S_c$ accounted for 61% of the difference between $F_c$ and $\widehat{ER}$, with the other 39% attributed to the advection components ($Av_c + Ah_c$). This indicates that

30   even on very flat terrain, the nocturnal advection term is significant.

**3.2 Inferred advection mechanisms**

Given that $Av_c + Ah_c$ is inferred, it is not possible to assess the relative contributions of the two components to the mass balance. However, gravity-induced drainage flow is a common mechanism for horizontal advective carbon losses. These flows are expected to be most readily identifiable from changes in $S_c$ because drainage flows are initiated under stable conditions when turbulence is suppressed. $F_c$ is expected to be small and $S_c$ correspondingly large (prior to the initiation of drainage flows) under these conditions.

Drainage flows are also expected to have a vertical spatial fingerprint. Specifically, their onset may result in changes in the magnitude of $S_c$ primarily in the lower layers of the control volume. At the study site, mean canopy height was $15.3 \pm 6.4$ m (SD; Table 1). Given that drainage flows generally confined to the trunk space below the canopy (Aubinet et al., 2003), and conservatively assuming that the canopy comprises the upper 30% of tree height, drainage flows may be confined to depths of <10 m, which are comparable to commonly reported values (Goulden et al., 2006; Mahrt et al., 2001). The impact on the carbon balance may nonetheless be large because as much as 70% of carbon is sourced from the soil in temperate forest ecosystems (Goulden et al., 1996a; Janssens et al., 2001; Law et al., 1999).

$S_c$ for the individual layers is presented in Figure 6. There was a clear decline in $S_c$ for all layers below 8 m when $u_*$ was less than approximately 0.25 m s$^{-1}$. In contrast, storage in the higher 8-16 m and 16-36 m layers continued to increase near linearly. We hypothesise that this indicates the onset of drainage flows at low levels under stable conditions, causing horizontal advective (i.e. $Ah_c$) losses of carbon from the lower layers of the control volume. The ongoing increases in storage in the 8-16 and 16-36 m layers may indicate that the carbon source in these layers originated primarily from the vegetation than from upward transfer from lower layers. In the interval between the $u_*$ thresholds for $F_c$ alone and $F_c + S_c$ (i.e. $0.32 \leq u_* \leq 0.42$ m s$^{-1}$), $Av_c + Ah_c$ was not significantly different to zero (see Figure 5). The linear relationship between each of the lower layers (0-0.5, 0.5-2, 2-4 and 4-8 m) and the mean 8-36 m layer in this interval $0.32 \leq u_* \leq 0.42$ m s$^{-1}$ can be used to extrapolate the expected rate of change for those layers in the absence of advection when $u_* < 0.31$ m s$^{-1}$.

Extrapolation of this linear relationship to conditions where $u^* < 0.32$ provides an estimate of the expected magnitude of $S_c$ in the absence of advection. If drainage flows are the primary advective mechanism, then the sum of $F_c$ and the linearly adjusted storage term should approximate $\widehat{ER}$. The correction to the storage in the 0-8 m layers when $u_* < 0.31$ m s$^{-1}$ increased the 0-36 m storage term such that for the interval $0 < u_* < 0.31$, the mass balance was approximately closed because $\widehat{ER} - F_c \approx S_c$ to within the uncertainty (95% confidence interval) of the bin means over this interval (Figure 7). This indicates that the decline in $S_c$ at lower layers was of approximately the same magnitude as the inferred advection, consistent with the presence of low-level drainage flows removing carbon from the control volume.

Our approach is subject to substantial uncertainty because the assumption that the linear relationship between levels holds for declining $u_*$ may not be correct. Direct observation of the proposed advective mechanism (terrain-induced drainage flows) would provide clarification of the mechanisms driving the nocturnal C dynamics at the site. While wind speed and direction were measured for all levels, the instrumentation lacked the resolution to detect the weak winds that generally characterise drainage flows in moderate terrain (typically less than 0.5 m/s; Aubinet et al., 2003; Goulden et al., 2006; Mahrt et al., 2001) (typically less than 0.5 m s$^{-1}$; Aubinet et al., 2003; Goulden et al., 2006; Mahrt et al., 2001). Nonetheless, the available data are consistent with a primary advective mechanism of terrain drainage flows.

**3.3 Effects of correction methods on diurnal and annual carbon balances**

The importance of the contribution of $S_c$ to the diurnal carbon balance is presented in Figure 8a. Given that the contribution averages approximately zero over the diurnal cycle, annual NEE sums for both $F_c$ and $F_c + S_c$ were comparable: approximately –450, –400 and –560 gC m$^{-2}$ a$^{-1}$ for 2012, 2013 and 2014, respectively (Table 4; small differences [< 20 gC m$^{-2}$ a$^{-1}$] were observed for $F_c$ versus $F_c + S_c$, reflecting small differences in parameters of functions used for gap filling). However, the diurnal dynamics were substantially changed with the addition of $S_c$, with increased amplitude and phase shift of peak carbon uptake (from midday – synchronous with the solar radiative peak – on average, to about 1100). Nocturnal $u_*$ dependency of $F_c + S_c$ indicates the presence of advection under weak turbulence. Following $u_*$ correction for this dependency (Figure 8b), estimated annual carbon uptake was reduced by 50–75 gC m$^{-2}$ a$^{-1}$.

Given that the majority of sites both internationally and in Australia do not have profile measurement systems, we discuss the effects of neglect of $S_c$ because this is the *de facto* approach taken for sites without profile systems. As a secondary option, a single point storage term (herein $S_{c\_pt}$) can be derived from the EC gas analyser. This will underestimate storage for taller towers, where much carbon accumulates within the control volume, and is subject to substantial error (Gu et al., 2012; Yang et al., 2007) but may nonetheless potentially reduce bias.

Summing the turbulent flux and single point estimate (i.e. $F_c + S_{c\_pt}$) and applying the $u_*$ correction, the diurnal cycle was relatively well-approximated on average (Figure 9a). $S_{c\_pt}$ sums to zero over each year, but substantially underestimates $S_c$ over the diurnal cycle (Figure 10). We expect to see a bias towards efflux because although the $u_*$ correction rectifies the nocturnal underestimation of efflux, the ventilation of accumulated carbon through the turbulent flux is not balanced by a decline of corresponding magnitude in storage. However, the results differed among years (Table 4). In 2012, $u_*$-corrected NEE from $F_c + S_{c\_pt}$ was comparable to the best estimate for $F_c + S_c$ (–375 and –382 gC m$^{-2}$ a$^{-1}$ for $F_c + S_{c\_pt}$ and $F_c + S_c$, respectively). In 2013 uptake was actually higher for $F_c + S_{c\_pt}$ (–362 vs –338 gC m$^{-2}$ a$^{-1}$ for $F_c + S_c$). In 2014, the value was substantially lower (–431 versus –480 gC m$^{-2}$ a$^{-1}$ for $F_c + S_c$).

The inconsistency of the annual NEE estimate between years arose due to the effect of error on model parameterisation and subsequent imputation. Random error estimates (derived from equation 8) are shown in Figure 11; at low flux magnitudes the error associated with $F_c + S_{c\_pt}$ was large relative to that for $F_c$ alone or $F_c + S_c$. This may be due to a mismatch in the timing of peak $S_{c\_pt}$ relative to $S_c$; small day-to-day variations in this timing may – in conjunction with noise in $S_{c\_pt}$ – result in large variations in NEE estimates for a given set of environmental conditions (as essentially demonstrated in Figure 11). Large errors in point-based storage estimates were also reported by Gu et al. (2012) for a forest ecosystem. This in turn affects the parameter estimates for $rb$ in the respiration function (equation 5) and α in the light response function (equation 6) in particular; even if this translates to small relative errors in GPP and ER, the effect on NEE - as the small difference between two large exchanges - may in relative terms be substantial.

Correction of $F_c$ yielded comparable nocturnal NEE to that estimated from $u_*$-corrected $F_c + S_c$ because it accounted for the nocturnal effects of storage and advection (Figure 9b). However, the morning increase in turbulent flux mixes nocturnally respired carbon stored in the control volume up through the measurement height. In the absence of a corresponding change in storage to account for this, the respiratory flux is counted twice, and annual NEE calculated using this method was underestimated by approximately 80 gC m$^{-2}$ a$^{-1}$ relative to the best estimate (Table 4). This is larger than the correction for $F_c + S_c$ associated with advection (50-75 gC m$^{-2}$ a$^{-1}$), which is consistent with the previously established fact that $S_c$ contributes more to $F_c$ underestimation than advection. In the absence of $S_c$ estimates, lower annual bias would be obtained for this study if the $u_*$ correction were *not* applied.

The double-counting problem might be mitigated at sites without storage measurements if, after nocturnal data correction, the morning 'flush' period, when the accumulated carbon in the control volume is vented following the re-establishment of thermally generated turbulence after sunrise, is identified and removed. This flush is expected to be manifested as an upward spike in $F_c$ following sunrise (e.g. Aubinet et al., 2012, Figure 5.2 therein). However, it may be difficult to identify objectively the flushing of accumulated $CO_2$ from the control volume from the behaviour of $F_c$ alone.

We observed no spike in $F_c$ indicating morning venting of accumulated carbon (Figure 8a). The magnitude of the expected $F_c$ spike depends on: (i) the quantity of carbon stored, which depends on prior respiration and advection, and; (ii) the relative timing and magnitude of increase of source / sink activity and turbulent activity. On average, $u_*$ reached $u_{*th}$ for $F_c$ approximately two hours after sunrise, whereas the storage term began to decline immediately when insolation increased (Figure 10). This indicates that carbon stored within the control volume began to be consumed by photosynthesis before efficient ventilation of the control volume was underway. The $CO_2$ mole fractions of the lowest levels measured by the profile system (0.5 and 2 m) also began to decline first (Figure 12), suggesting that shrubs reach the light-compensation point earlier after sunrise than do trees probably because lower near-surface temperatures are expected to suppress respiration. The effect on storage was small since the layers represent a small proportion of the control volume.

Early-morning photosynthesis may be substantial in eucalypt-dominated ecosystems in which the characteristically pendulous (in some species up to 75% of mature leaves typically hang at angles > 80 degrees from horizontal; Pereira et al., 1987), amphistomatous leaves evolved to maximise incident radiation at low sun angles, which shifts photosynthetic activity towards periods with lower vapour pressure deficit (James and Bell, 1996). Mutual shading would partially counteract this effect at low sun angles, but this may have less effect in systems with sparse canopies such as the woodland in this study. Similarly, an Australian temperate eucalypt forest site with long-term turbulent flux and $CO_2$ profile measurements showed no morning spike in $CO_2$ efflux (Van Gorsel et al., 2007, Figure 4 therein).

Alternatively, the flush period may be defined as occurring between sunrise and the time at which the quasi-instantaneous, vertically integrated control volume mean of the $CO_2$ mole fraction approaches its long-term temporal mean. The long term temporal mean in this study was 395 ppm $CO_2$, within 2 ppm of the mean southern-hemisphere ambient $CO_2$ mole fraction during 2012-2014. As long as the $CO_2$ mole fraction in the control volume exceeds that of the overlying atmosphere, there is potential for flushing of previously respired carbon from the control volume. If it is assumed that, over a sufficient period, mean $CO_2$ mole fraction in the control volume is in equilibrium with the overlying atmosphere, then the concentration gradient between the control volume and the overlying atmosphere will continue to facilitate upward turbulent transfer of $CO_2$ until the control volume $CO_2$ mole fraction declines to its mean. In this study, this occurred on average at approximately 0900 local standard time compared with austral / autumnal vernal equinox sunrise times of 0622 and 0611, respectively. Ensuring that no data were included during the period where flushing of nocturnally respired carbon could potentially have occurred would require removal of data between 0600-0900.

Essentially the reverse process to that described above occurred during daytime and early evening. Negative daytime $S_c$ after 0900 reflects ongoing photosynthetic $CO_2$ drawdown within the control volume. $S_c$ approached zero at approximately 1600 on average, because insolation, which drives photosynthesis, is out of phase with surface heating, which drives $u_*$ and respiration and so photosynthesis declines earlier than respiration and turbulent mixing. Therefore, NEE estimates continued to be biased throughout the day in the absence of $S_c$. However, this has little effect on long-term carbon balance because the daytime bias in $F_c$ due to neglect of $S_c$ is offset by nocturnal bias of opposite sign in the early evening. Given that the $CO_2$ mole fraction dropped below the mean at 0900 and remained below it until 2100, $S_c$ summed over this period must be approximately zero.

The period between sunset (austral / autumnal vernal equinox sunset times: 1830 and 1813) and 2100 is therefore the inverse of the flush period between 0600-0900, with the $CO_2$ mole fraction in the control volume *lower* than ambient which, in the absence of respiratory carbon production, would facilitate downward turbulent transfer of carbon. The increase in the storage term after sunset (Figure 10) partly reflects the 'refilling' of the control volume reservoir following daytime

drawdown, which may contribute to the observation that nocturnal $S_c$ reached its highest value when $u_* > u_{*th}$. While the effect on annual NEE may be minor, it violates the assumption in the $u_*$-correction approach that only the turbulent flux term is significant when $u_* > u_{*th}$. The implications of this for sites without storage measurements are discussed below.

5    **3.4 Effects of neglecting carbon storage on physiological interpretation of data**

From the perspective of deriving annual NEE sums, it is daytime rather than nocturnal measurements that are more critical; applying the $u_*$ correction to either $F_c$ or $F_c + S_c$ resulted in similar estimates of nocturnal NEE on average. But the effects of neglecting $S_c$ depend on time of night. $F_c$ underestimated NEE following sunset, even where $u_* > u_{*th}$ for $F_c$.

10   However, a secondary nocturnal problem is recognised in the literature: when $u_*$ increases following extended calm periods, stored $CO_2$ is vented from the control volume, which artificially inflates $F_c$ relative to the true source term, the extent of which effect will depend on the importance of advection (Aubinet et al., 2012). When a $u_*$ threshold is imposed, such periods are likely to be included in the retained data. This has the opposite effect to the early-evening effect, and is more likely to be problematic later in the evening when stable stratification and substantial storage of respired carbon is more
15   likely. Both effects were observed in the nocturnal progression of $S_c$ for periods when $u_* > u_{*th}$ (Figure 13): $S_c$ was on average $> 0$ for the first 4-6 hours after sunset and $< 0$ afterwards. On balance, the effect in this study was to increase slightly the estimation of ER. This explains why $S_c$ was slightly positive when $u_* > u_{*th}$ in Figure 3, and why $F_c$ was slightly lower than $F_c + S_c$ at night (primarily in the early evening) in Figure 9b.

20   However, given that temperature decreases over the evening, this suggests that the slope of temperature response functions will be slightly increased for $F_c + S_c$ versus $F_c$ alone. Given that the optimisation procedure minimises the prediction error, this may not have a large quantitative effect averaged over the evening, but interpretation of system response to temperature is distorted. Moreover, extrapolation beyond the parameterisation domain (e.g. estimation of daytime ER) may result in substantial error because distortion of function parameters (e.g. $E_o$ and $rb$ in equation 5) will potentially result in systematic
25   error (because the function optimised using $F_c$ will underestimate NEE at high temperatures). Any bias in estimated daytime ER will then necessarily propagate to estimation of GPP (commonly calculated as NEE – ER). Because these errors are offsetting, this is not likely to have a large effect on annual NEE estimates.

Similar distortion of response to insolation occurs during the day. The addition of $S_c$ substantially affects diurnal NEE
30   dynamics, particularly during the morning, which affects the interpretation of the controls on NEE. For example, Figure 14 shows the difference in radiation use efficiency (RUE – here simply defined as the ratio of mean NEE to mean insolation) during daylight hours for $F_c$ alone versus $F_c + S_c$. RUE was higher in the early morning, and declined more sharply, when

NEE $= F_c + S_c$. Such declines in RUE are often associated with stomatal response to increasing VPD, and so the importance of this driver may be missed or minimised when $S_c$ is not measured.

Application of light-response function analysis to daytime data to extract either photosynthetic or respiratory parameters is problematic in the absence of storage measurements because $F_c \neq$ NEE during most of the day. The estimation of ER (and quantum efficiency) derived from light response function analysis (e.g. Gilmanov et al., 2003; Lasslop et al., 2010) is strongly dependent on the magnitude of observed NEE when insolation is low (sunrise and sunset), and thus the effect of neglecting the storage term may be particularly distorting to these parameters (Aubinet et al., 2012).

**3.5 Sources of uncertainty**

One of the largest sources of uncertainty in annual NEE estimates is expected to derive from uncertainty in $u_{*th}$ (Barr et al., 2013; Papale, 2006). We propagated this uncertainty to annual NEE (for both $F_c$ and $F_c + S_c$) by filtering and gap-filling the data using the lower and upper bounds of the 95% confidence interval (CI) of the normally distributed population ($N = 10^3$) of $u_{*th}$ derived from CPD (Table 3). Much larger effects were evident for the lower uncertainty bound ($\mu - 2\sigma$, where $\mu$ is the best estimate for $u_{*th}$ and $\sigma$ is the standard deviation), which is to be expected because systematic errors in nocturnal flux measurement occur at low $u_*$. However, there should be no systematic variation in NEE when $u_* > u_{*th}$. The direct effect of the upper uncertainty bound ($\mu + 2\sigma$) is expected to be minimal. While the reduction of nocturnal data availability for higher $u_{*th}$ is expected to increase parameter imputation uncertainty, the effect here was minor, with annual NEE for $u_{*th} = \mu$ and $u_{*th} = \mu + 2\sigma$ differing by <10 gC m$^{-2}$ a$^{-1}$ in all years.

The uncertainty in $u_{*th}$ was greater for $F_c + S_c$ than for $F_c$ alone due to the additional random error inherent in $S_c$ (see Finnigan, 2006 for further discussion), which feeds into the change-point detection process. This propagated to larger uncertainty in the lower bound for annual NEE (50-75 gC m$^{-2}$ a$^{-1}$ compared to 20-40 gC m$^{-2}$ a$^{-1}$ for $F_c$), despite the effect of $u_*$ correction for $F_c$ was almost double that for $F_c + S_c$. This is because for $F_c + S_c$, the lower bound of the $u_{*th}$ uncertainty is below the 1$^{st}$ percentile of the nocturnal data, such that the full effect of advection was propagated to annual NEE but for $F_c$ alone, it was closer to the 40$^{th}$ percentile, such that only a small proportion of storage and advection occurred in the interval between $u_{*th} = \mu$ and $u_{*th} = \mu - 2\sigma$.

While the lower bound uncertainty due to $u_*$ threshold determination error was therefore of a magnitude only slightly smaller than the bias introduced by applying the $u_*$ correction in the absence of storage measurements, the uncertainty resulted in an increase in the potential uptake of carbon, whereas the bias was of opposite sign. Thus the uncertainty ranges for $F_c$ and for $F_c + S_c$ do not overlap. In this sense, the uncertainty range for $F_c$ alone is meaningful only in a very narrow sense (i.e. in terms of what has been measured as opposed to true source / sink uncertainty), and should not be reported as determining the

uncertainty in annual NEE. We have determined that the annual NEE estimate derived from $F_c + S_c$ is not contained within the $u_{*th}$ uncertainty interval for annual NEE derived from $F_c$ alone.

It should also be noted that the large lower-bound uncertainty in annual NEE derived from $F_c + S_c$ is very likely overestimated. If the lack of correlation between $u_*$ and $F_c$ above $u_{*th}$ indicates that additional terms in the mass balance are negligible, then $S_c$ should approach zero at $u_{*th}$. Given that this is what we observed (Figure 3), and the measurement system (and associated measurement errors) for $S_c$ is independent of that for $F_c$, this is an independent validation of $u_{*th}$. It is not clear how such information might be used in the context of frequentist statistical analysis, but it strongly suggests that the uncertainty bounds for NEE that include the effects of $u_{*th}$ uncertainty presented here are unrealistically large.

The NEE uncertainty contribution of combined random and model error was small ($<25$ gC m$^{-2}$ a$^{-1}$) by comparison (Table 3), $< 10\%$ of annual NEE for each year. Model error was greater than random uncertainty for both day and night, although the difference was larger at night. This is because the majority of nocturnal observational data was removed by the $u_*$ correction (thereby reducing random error) and replaced with model data. Given that the signal:noise is lower at night, model error is expected to be relatively large. The method used calculates and compounds observation – model data differences, but the observational data already contain random error so this necessarily inflates the propagated model uncertainty above errors associated with missing driver information or systematic measurement error. This problem is partially propagated to the daytime because random error contributes to uncertainty in the nocturnally derived parameters of equation 5, which are then used to calculate the ER component of daytime NEE (equation 6). Moreover, the light-response function estimates are affected by daytime random error, which explains why model error was relatively large comparable to nocturnal values during the day despite 70-90% of the data being retained.

Annual NEE uncertainty due to random and model error was approximately 20% greater for $F_c + S_c$ than for $F_c$ alone, due to additional random error to that for $F_c$ arising from the addition of $S_c$ (Figure 11). This is largely nocturnally determined because the storage term is smaller and less variable during the daytime when fluxes are largest. The increased annual uncertainty of $F_c + S_c$ is largely due to higher model uncertainty, which most likely reflects the propagation of random error to model uncertainty through effects on non-linear parameter estimation.

The interdependence of model and random error technically renders invalid the assumption of independence in equation 9. However, the effect is to increase rather than to decrease uncertainty, which is small. While there are methods to separate model and random components (see Dragoni et al., 2007), this generally requires co-location of two instrument arrays. However, the daily differencing procedure we used is known to overestimate error by up to a factor of 2 (Billesbach, 2011; Dragoni et al., 2007) due both to potential wind-dependent temporal variations in source / sink strength (which may

materially affect annual NEE estimates; Griebel et al., 2016) and because some signal is included in the differencing procedure.

It should be emphasised that there are numerous sources of uncertainty that have not been quantified here. Perhaps most important of these is systematic errors in the measurements themselves, which may be an extremely important source of true uncertainty (Lasslop et al., 2008). Thus the uncertainties reported here for $F_c + S_c$ also should not be formally interpreted as total uncertainty in the true source / sink term, but as the uncertainty contributed by a subset of quantifiable errors.

**34 Conclusions**

We used a simple method to infer advection from measurements combined with a simple and widely used empirical respiration model. Even at our very flat site, approximately 40% of flux underestimation was attributable to advection. Observation of reductions in storage at lower levels (within 8m of the surface) in response to declining $u_*$ indicate that the most likely advective mechanism is terrain drainage flows. High resolution measurements of wind directions within the control volume would be invaluable for directly detecting the presence of terrain-aligned flows, and are planned for this site.

Given that this site is relatively ideal, this follows earlier findings that drainage-induced advection is likely to affect most sites to some degree. Where the NEE time series consists of turbulent flux and storage, nocturnal correction for these effects should be made under all circumstances. The level of bias incurred in the absence of storage measurements is contingent on the contribution of advection to the nocturnal mass balance. Nocturnal $u_*$ correction is not advisable if the dominant term in nocturnal flux underestimation is expected to be storage (as found in this study), because the biasing effect of correcting the nocturnal component without the counterbalancing daytime component is larger than correcting for the smaller advection term. At sites with severe advection problems, neglect of storage is relatively less important, because if $CO_2$ is drained away nocturnally, the early-morning venting that causes bias following nocturnal correction is small.

But this contingency underscores the intractable nature of the problem: the relative contributions of storage and advection to the nocturnal mass balance cannot be quantitatively assessed in the absence of profile measurements. Moreover, even at sites where drainage flows are known to regularly occur at night, it is likely that shear-induced turbulence penetrates below canopy only under strong winds; the rarity of such conditions may result in the rejection of an unacceptably large number of data. Where this is the case, profile measurements are required to increase the proportion of available nocturnal data because storage increases prior to the onset of drainage flows, which only occur once the cooling air mass adjacent to the surface achieves sufficient density to overcome friction and begin to flow (Van Gorsel et al., 2007).

Storage measurements nonetheless introduce some complications for data interpretation. The additional random error in nocturnal storage measurements increases uncertainty in $u_*$ threshold and, correspondingly, annual NEE (uncertainties due to direct random observation error and imputation error were small by comparison). But as we have argued, the lower bound uncertainty for $u_*$ threshold is unrealistic, since the storage term on average approaches zero at the $u_*$ threshold. This behaviour is expected if the central $u_*$ threshold estimate from change point detection is approximately correct. Even if these uncertainties are considered accurate, when propagated to annual NEE, the resulting uncertainty intervals for the sum of turbulent flux and storage versus turbulent flux alone do not overlap. This indicates that biases are not subsumed within (quantified) uncertainties; effectively, profile measurements reduce precision and increase accuracy of annual NEE estimates.

We therefore believe that for both OzFlux and Fluxnet, the installation of profile systems for sites with trees (woodlands, forests, savannas) is extremely important to ensure that both determination of annual carbon exchange and interpretation of ecosystem processes are accurate. At the very least, the issues explored here need to be taken into consideration during data analysis, and alternative methods of estimating uncertainties at sites without profile systems need to be developed. For sites under the auspices of the Integrated Carbon Observation System (ICOS: www.icos-ri.eu), profile systems are mandatory; while this is not yet the case for OzFlux and Fluxnet, for accurate estimates of annual NEE, profile systems are vital.

We used a simple method to infer advection from measurements combined with a simple and widely used empirical respiration model. Even at our very flat site, approximately 40% of flux underestimation was attributable to advection, which was consistent with previous findings that advection is likely to be important at most sites (Aubinet, 2008). While the mechanism for advection could not be confirmed, the behaviour of the profile measurements suggested horizontal advection associated with from landscape drainage flows. High resolution measurements of wind directions within the control volume would be invaluable for directly detecting the presence of terrain-aligned flows. Such measurements are planned for this site.

We demonstrated the importance of profile measurements for improved measurements of annual carbon balances. The nocturnal biases introduced by the neglect of storage measurements altogether were up to 25% of annual NEE at our study site and were larger than the actual correction applied when storage measurements were included, so that less bias would ensue if the correction were not applied. Use of the point-based storage estimate caused large errors in parameter estimation and thus inconsistent effects on gap-filled annual sums of NEE.

The level of bias from neglecting storage is likely to depend on the site, contingent on the importance of advection. At sites without profile measurements, nocturnal $u_*$ correction is not advisable if the dominant term in nocturnal flux underestimation is expected to be storage because the effect of correcting the nocturnal component without the counterbalancing daytime component is larger than correcting for the smaller advection term. At sites with severe advection

problems, nocturnal correction is strongly recommended, and neglect of profile measurements is likely to be relatively less important. This is because if $CO_2$ is drained away nocturnally, the early-morning venting that causes bias following nocturnal correction is small.

This contingency underscores the intractable nature of the problem: the relative contributions of storage and advection to the nocturnal mass balance cannot be quantitatively assessed in the absence of profile measurements. Moreover, even at sites where drainage flows are known to regularly occur at night, it is likely that shear-induced turbulence penetrates below canopy only under strong winds; the rarity of such conditions may result in the rejection of an unacceptably large number of data. Where this is the case, profile measurements are required to increase the proportion of available nocturnal data because storage increases prior to the onset of drainage flows, which only occur once the cooling air mass adjacent to the surface achieves sufficient density to overcome friction and begin to flow (Van Gorsel et al., 2007).

Storage measurements introduce some complications for data interpretation. The additional random error inherent in nocturnal storage measurements meant that the lower uncertainty bound for $u_{*th}$ was approximately zero for each year and, as such, the full effect of advection was included in the associated annual NEE estimate. However, the profile system provides an independent check of the $u_{*th}$, which, in this case, indicates the central estimate of $u_{*th}$ is correct and the lower bound of the uncertainty range is unrealistic. By comparison to uncertainty in $u_{*th}$, random and model error uncertainties in annual NEE were small, and despite the storage term increasing the error induced by approximately 25%, in absolute terms the combined uncertainty was < 10% of annual NEE. Moreover, the reduced annual uncertainty interval obtained when storage was excluded was largely meaningless because it did not contain the actual estimate of annual NEE when storage was included.

[revised manuscript text omitted]

5    Figure 10: mean diurnal cycle of aggregate and component $S_c$ (LH axis; horizontal solid line represents mean $S_c$) and $u*$ (RH axis; horizontal dashed line represents change point-derived nocturnal $u*$ threshold);  note that day length as indicated by insolation is slightly greater than 12 hours  due to missing data for several months during winter / spring 2013, thereby slightly biasing the data towards longer photoperiod.

[Figure]

**Figure 11: standard deviation of estimated random error (σ[δ]) as a function of flux magnitude for turbulent flux (F$_c$), turbulent flux plus profile-based storage estimate (F$_c$ + S$_c$) and turbulent flux plus point-based storage estimate (Fc + S$_{c\_pt}$).**

[Figure]

Figure 12: mean diurnal cycle of $CO_2$ mole fraction measured by profile system for given layers (coloured lines; grey line represents 0-36m height-integrated control volume mean) and by eddy covariance infra-red gas analyser at 36m (black line). Note that due to minor discrepancies between instruments, mole fractions for the eddy covariance IRGA are here baselined to the height-integrated profile IRGA multi-annual mean (394.6ppm).

[Figure]

**Figure 13: dependence of $S_c$ (including only data where $u_* > u_{*th}$) on time after sunset (upper panel; dotted line is air temperature); cumulative percentage of total nocturnal $S_c$ observations (lower panel).**

[Figure]

**Figure 14: LH axis) effects of addition of $F_c$ to $S_c$ on radiation use efficiency (RUE); RH axis) $S_c$ as a proportion of $F_c$.**

**Table 1: site characteristics**

| | |
|---|---|
| Latitude, longitude (º dec.) | -36.673215, 145.029247 |
| Slope (º) | <1 |
| Aspect | N/A |
| Dominant overstorey (>90%*) species | *Eucalyptus microcarpa* |
| Dominant understorey (>90%*) species | *Cassinia arculeata* |
| Mean canopy height ±SD (m) | 15.3±6.4 |
| Leaf area index ($m^2$ $m^{-2}$) | ~1.1 |
| Mean annual temperature (ºC) | 15.9 |
| Mean annual precipitation (mm)**: long term (1971-2000) | 560 |

\* By biomass (although also by number of individuals in the case of the overstorey)

\*\* From nearest long-term rainfall measurement site (Mangalore Airport; Bureau of Meteorology station ID 088109)

**Table 2: site instrumentation**

| Measurement | Instrument | Manufacturer |
|---|---|---|
| Wind vectors / virtual temperature | CSAT3 | Campbell Scientific Instruments |
| Radiation components | CNR4 | Kipp and Zonen |
| $CO_2$ mole fraction (eddy covariance) | LI7500 | Licor Biosciences |
| $CO_2$ mole fraction (profile) | LI840 | Licor Biosciences |
| Temperature / humidity | HMP45C | Vaisala |
| Wind speed / direction (profile) | Wind Sentry Set | RM Young |
| Barometric pressure | PTB110 | Vaisala |
| Volumetric soil water content | CS616 | Campbell Scientific Instruments |
| Soil heat flux | HFP01 | Hukseflux |
| Soil temperature | TCAV | Campbell Scientific Instruments |
| Data logging | CR3000 | Campbell Scientific Instruments |

Table 3: lower 95%CI bound (μ - 2σ), mean (μ), and upper 95%CI bound (μ + 2σ) of Gaussian PDF of $u_{*th}$ (derived from change point detection of bootstrapped samples - see Methods), data percentile (i.e. percentage data excluded for each $u_{*th}$) and resulting imputed annual estimate of NEE. Note: i) μ – 2σ set to zero if < 0 (e.g. $F_c + S_c$ in 2013); ii) respiration and light response function analysis could not find a solution for $F_c + S_c$ in 2013 when $u_* = 0.73$ (insufficient data for robust statistical fit).

| Year | Condition | $F_c$ | | | $F_c + S_c$ | | |
|------|-----------|-------|---|---|-------------|---|---|
| | | $u_*$ (m s$^{-1}$) | Data percentile | Annual NEE (gC m$^{-2}$ a$^{-1}$) | $u_*$ (±95%CI) | Data percentile | Annual NEE (gC m$^{-2}$ a$^{-1}$) |
| 2012 | μ - 2σ | 0.26 | 47 | -323.1 | 0.01 | <1 | -450.6 |
| | μ | 0.39 | 60 | -299.0 | 0.30 | 50 | -381.5 |
| | μ + 2σ | 0.52 | 72 | -304.7 | 0.59 | 77 | -387.1 |
| 2013 | μ - 2σ | 0.19 | 38 | -285.2 | 0 | 0 | -364.1 |
| | μ | 0.40 | 61 | -250.4 | 0.32 | 53 | -313.3 |
| | μ + 2σ | 0.61 | 79 | -251.4 | 0.73 | 87 | - |
| 2014 | μ - 2σ | 0.23 | 43 | -433.4 | 0.02 | <1 | -552.7 |
| | μ | 0.42 | 63 | -395.2 | 0.32 | 53 | -483.2 |
| | μ + 2σ | 0.61 | 79 | -398.8 | 0.62 | 80 | -481.9 |

Table 4: gap-filled annual NEE (gC m$^{-2}$ a$^{-1}$) for 2012-2014 obtained following different data treatment ($F_c$: turbulent flux only; $F_{c\_u*\_corr}$: turbulent flux with low $u_*$ conditions removed; $F_c + S_{c\_pt}$: summed turbulent flux and point-based storage estimate; $(F_c + S_{c\_pt})_{u*\_corr}$: summed turbulent flux and point-based storage estimate with low $u_*$ conditions removed; $F_c\_S_c$: summed turbulent flux and profile-based storage estimate; $(F_c + S_c)_{u*\_corr}$: summed turbulent flux and profile-based storage estimate with low $u_*$ conditions removed).

| Year | $F_c$ | $F_{c\_u*\_corr}$ | $F_c + S_{c\_pt}$ | $(F_c + S_{c\_pt})_{u*\_corr}$ | $F_c + S_c$ | $(F_c + S_c)_{u*\_corr}$ |
|------|-------|-------------------|-------------------|--------------------------------|-------------|--------------------------|
| 2012 | -463 | -301 | -489 | -375 | -447 | -382 |
| 2013 | -403 | -267 | -492 | -362 | -388 | -338 |
| 2014 | -572 | -393 | -586 | -431 | -551 | -480 |